# An orphan $cbb_3$-type cytochrome oxidase subunit supports *Pseudomonas aeruginosa* biofilm growth and virulence

Jeanyoung Jo, Krista L Cortez, William Cole Cornell, Alexa Price-Whelan, Lars EP Dietrich*

Department of Biological Sciences, Columbia University, New York, United States

**Abstract** Hypoxia is a common challenge faced by bacteria during associations with hosts due in part to the formation of densely packed communities (biofilms). $cbb_3$-type cytochrome $c$ oxidases, which catalyze the terminal step in respiration and have a high affinity for oxygen, have been linked to bacterial pathogenesis. The pseudomonads are unusual in that they often contain multiple full and partial (i.e. 'orphan') operons for $cbb_3$-type oxidases and oxidase subunits. Here, we describe a unique role for the orphan catalytic subunit CcoN4 in colony biofilm development and respiration in the opportunistic pathogen *Pseudomonas aeruginosa* PA14. We also show that CcoN4 contributes to the reduction of phenazines, antibiotics that support redox balancing for cells in biofilms, and to virulence in a *Caenorhabditis elegans* model of infection. These results highlight the relevance of the colony biofilm model to pathogenicity and underscore the potential of $cbb_3$-type oxidases as therapeutic targets.

DOI: https://doi.org/10.7554/eLife.30205.001

## Introduction

Among the oxidants available for biological reduction, molecular oxygen ($O_2$) provides the highest free energy yield. Since the accumulation of $O_2$ in the atmosphere between ~2.4 and 0.54 billion years ago (*Kirschvink and Kopp, 2008*; *Dietrich et al., 2006b*), organisms that can use it for growth and survival, and tolerate its harmful byproducts, have evolved to exploit this energy and increased in complexity (*Knoll and Sperling, 2014*; *Falkowski, 2006*). At small scales and in crowded environments, rapid consumption of $O_2$ leads to competition for this resource and has promoted diversification of bacterial and archaeal mechanisms for $O_2$ reduction that has not occurred in eukaryotes (*Brochier-Armanet et al., 2009*). The various enzymes that allow bacteria to respire $O_2$ exhibit a range of affinities and proton-pumping efficiencies and likely contribute to competitive success in hypoxic niches (*Morris and Schmidt, 2013*). Such environments include the tissues of animal and plant hosts that are colonized by bacteria of high agricultural (*Preisig et al., 1996*) and clinical (*Way et al., 1999*; *Weingarten et al., 2008*) significance.

The opportunistic pathogen *Pseudomonas aeruginosa*, a colonizer of both plant and animal hosts (*Rahme et al., 1995*), has a branched respiratory chain with the potential to reduce $O_2$ to water using five canonical terminal oxidase complexes: two quinol oxidases ($bo_3$ (Cyo) and a $bd$-type cyanide-insensitive oxidase (CIO)) and three cytochrome $c$ oxidases ($aa_3$ (Cox), $cbb_3$-1 (Cco1), and $cbb_3$-2 (Cco2)) (*Figure 1A*). Several key publications have described *P. aeruginosa*'s complement of terminal oxidases and oxidase subunits, revealing features specific to this organism (*Williams et al., 2007*; *Comolli and Donohue, 2004*; *Alvarez-Ortega and Harwood, 2007*; *Arai et al., 2014*; *Kawakami et al., 2010*; *Jo et al., 2014*). *P. aeruginosa* is unusual in that it encodes two oxidases belonging to the $cbb_3$-type family. These enzymes are notable for their relatively high catalytic activity at low $O_2$ concentrations and restriction to the bacterial domain (*Brochier-Armanet et al., 2009*;

*For correspondence:
LDietrich@columbia.edu

Competing interests: The authors declare that no competing interests exist.

**eLife digest** Bacteria often form communities called biofilms to make them stronger and more 'invincible'. However, when these communities become too crowded, oxygen levels can drop, which makes it harder for them to survive. Some types of bacteria, such as *Pseudomonas aeruginosa*, have found different ways to cope with lower levels of oxygen. For example, they produce enzymes that use oxygen more efficiently or are better at scavenging low concentrations of oxygen.

When organisms – including bacteria – produce energy, they break down nutrients into small molecules to extract electrons. These electrons are then transported along their membrane until they reach their final destination – an oxygen molecule. Studies of *P. aeruginosa* grown in the laboratory have shown that it uses several types of enzymes called terminal oxidases to complete this last electron transfer. The bacterium can also make chemicals that help to shuttle electrons to remote oxygen sources. For example, they can produce compounds called phenazines that can transport electrons and also compensate for low oxygen levels.

However, the conditions in biofilms can be very different to those in a laboratory environment, and until now it was not known what role the different oxidases play in biofilm communities, or how phenazines can compensate for low oxygen levels.

To investigate this further, Jo et al. studied *P. aeruginosa* in an artificial biofilm environment and in a nematode worm host. The results showed that a specific part of the terminal oxidases – a protein called CcoN4 – was necessary for *P. aeruginosa* to grow optimally in both instances. Mutant bacteria that lacked CcoN4 struggled to survive. Moreover, bacteria containing CcoN4 were able to deliver the electrons to phenazines. This suggests that CcoN4 is also needed for phenazines to work properly.

This study shows that blocking terminal oxidases that contain CcoN4 can weaken *P. aeruginosa* and consequently its ability to cause infections. Furthermore, these types of terminal oxidases are only found in bacteria, which makes them attractive targets for potential drugs that would have minimal side effects on the host's metabolism. *P. aeruginosa* infections are a leading cause of death for people suffering from cystic fibrosis, a genetic condition that affects the lungs and the digestive system. A better understanding of what makes *P. aeruginosa* so infectious will help to find new treatments for these patients.

DOI: https://doi.org/10.7554/eLife.30205.002

*Pitcher and Watmough, 2004*). (The *P. aeruginosa* $cbb_3$-type oxidases are often referred to as $cbb_3$-1 and $cbb_3$-2; however, we will use 'Cco1' and 'Cco2' for these enzymes, consistent with the annotations of their encoding genes.) Most bacterial genomes that encode $cbb_3$-type oxidases contain only one operon for such a complex, which is induced specifically under conditions of $O_2$ limitation (*Cosseau and Batut, 2004*). In *P. aeruginosa*, the *cco2* operon is induced during growth at low $O_2$ concentrations, but the *cco1* operon is expressed constitutively at high levels (*Comolli and Donohue, 2004*; *Kawakami et al., 2010*).

An additional quirk of the *P. aeruginosa* terminal oxidase complement lies in the presence of genes for 'orphan' $cbb_3$-type subunits at chromosomal locations distinct from the *cco1* and *cco2* operons. While the *cco1* and *cco2* operons, which are chromosomally adjacent, each contain four genes encoding a functional Cco complex (consisting of subunits N, O, P, and Q), the two additional partial operons *ccoN3Q3* and *ccoN4Q4* each contain homologs coding for only the Q and catalytic N subunits (*Figure 1B*). Expression of the *ccoN3Q3* operon is induced under anaerobic denitrification conditions (*Alvarez-Ortega and Harwood, 2007*), and by nitrite exposure during growth under 2% $O_2$ (*Hirai et al., 2016*). During aerobic growth in liquid cultures, *ccoN4Q4* is induced by cyanide, which is produced in stationary phase (*Hirai et al., 2016*). However, additional expression studies indicate that *ccoN4Q4* transcription is influenced by redox conditions, as this operon is induced by $O_2$ limitation (*Alvarez-Ortega and Harwood, 2007*) and slightly downregulated in response to pyocyanin, a redox-active antibiotic produced by *P. aeruginosa* (*Dietrich et al., 2006a*).

In a recent study, Hirai *et al.* characterized the biochemical properties and physiological roles of *P. aeruginosa* $cbb_3$ isoforms containing combinations of canonical and orphan subunits (*Hirai et al., 2016*). In a strain lacking all of the aerobic terminal oxidases, expression of any isoform conferred

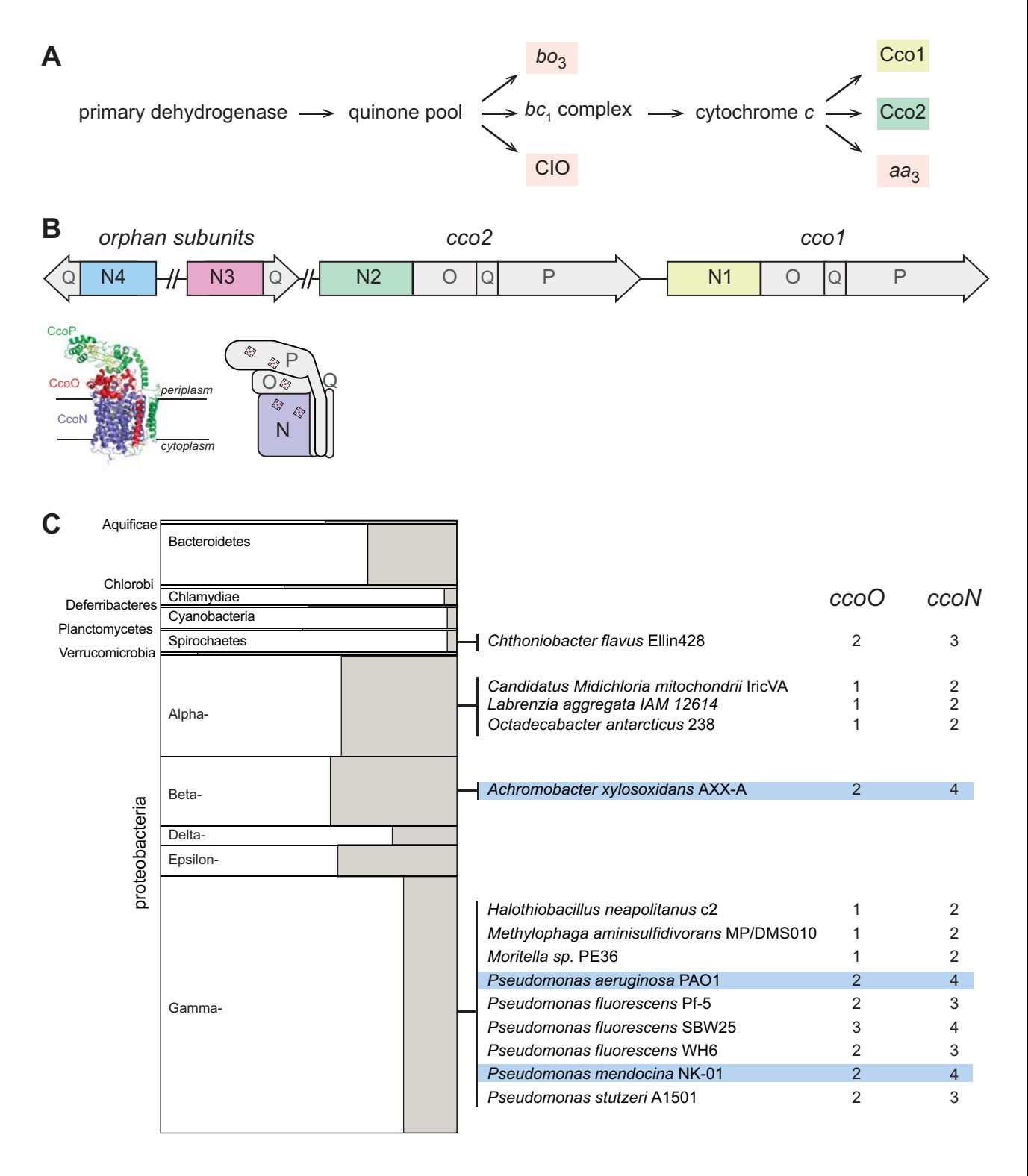

**Figure 1.** The respiratory chain and arrangement of *cco* genes and protein products in *P. aeruginosa*, and the phylogenetic distribution of orphan *ccoN* genes. (**A**) Branched electron transport chain in *P. aeruginosa*, containing five canonical terminal oxidases. (**B**) Organization of *cco* genes in the *P. aeruginosa* genome. The cartoon of the Cco complex is based on the Cco structure from *P. stutzeri* (PDB: 3mk7) (**Buschmann et al., 2010**). (**C**) Left: graphical representation of the portion of genomes in each bacterial phylum that contain *ccoO* and *N* homologs. The clades Chrysiogenetes, Gemmatimonadetes, and Zetaproteobacteria were omitted because they each contain only one species with *ccoO* and *N* homologs. The height of

*Figure 1 continued on next page*

*Figure 1 continued*

each rectangle indicates the total number of genomes included in the analysis. The width of each shaded rectangle represents the portion of genomes that contain *ccoN* homologs. Middle: genomes that contain more *ccoN* than *ccoO* homologs (indicating the presence of orphan *ccoN* genes) are listed. Right: numbers of *ccoO* and *ccoN* homologs in each genome. Blue highlights genomes containing more than one orphan *ccoN* homolog.

DOI: https://doi.org/10.7554/eLife.30205.003

the ability to grow using $O_2$, confirming that isoforms containing the orphan N subunits are functional. When preparations from wild-type, stationary-phase *P. aeruginosa* cells were separated on 2D gels and probed with anti-CcoN4 antibody, this subunit was detected at the same position as the assembled CcoNOP complex, showing that CcoN4-containing heterocomplexes form in vivo. Furthermore, the authors found that the products of *ccoN3Q3* and *ccoN4Q4* contributed resistance to nitrite and cyanide, respectively, during growth in liquid cultures under low-$O_2$ conditions. While these results provide insight into contributions of the $cbb_3$ heterocomplexes to growth in liquid cultures, potential roles for N3- and N4-containing isoforms in biofilm growth and pathogenicity have yet to be explored.

The biofilm lifestyle—in which cells grow in a dense community encased in a self-produced matrix—has been linked to the establishment and persistence of infections in diverse systems (*Edwards and Kjellerup, 2012*; *Rybtke et al., 2015*). Biofilm development promotes the formation of $O_2$ gradients such that cells at a distance from the biofilm surface are subjected to hypoxic or anoxic conditions (*Werner et al., 2004*). Using a colony morphology assay to study redox metabolism and its relationship to community behavior, we have shown that $O_2$ limitation for cells in biofilms leads to an imbalance in the intracellular redox state. This can be relieved by a change in community morphology, which increases the surface area-to-volume ratio of the biofilm and therefore access to $O_2$ for resident cells (*Kempes et al., 2014*). For *P. aeruginosa* cells in biofilms, the intracellular accumulation of reducing power can also be prevented by production and reduction of endogenous antibiotics called phenazines, which mediate extracellular electron transfer to oxidants available at a distance (*Dietrich et al., 2013*). We have found that biofilm-specific phenazine production contributes to pathogenicity in a murine model of acute pulmonary infection (*Recinos et al., 2012*), further illustrating the importance of phenazine-mediated redox balancing for *P. aeruginosa* cells in communities.

Because of the formation of an $O_2$ gradient inherent to the biofilm lifestyle, we hypothesized that the differential regulation of the *P. aeruginosa cco* operons affects their contributions to metabolic electron flow in biofilm subzones. We evaluated the roles of various $cbb_3$-type oxidase isoforms in multicellular behavior and virulence. Our results indicate that isoforms containing the orphan subunit CcoN4 can support survival in biofilms via $O_2$ and phenazine reduction and contribute to *P. aeruginosa* pathogenicity in a *Caenorhabditis elegans* 'slow killing' model of infection.

## Results

### A small minority of bacterial genomes encode $cbb_3$-type oxidase subunits in partial ('orphan') operons

Biochemical, genetic, and genomic analyses suggest that the CcoN and CcoO subunits, typically encoded by an operon, form the minimal functional unit of $cbb_3$-type oxidases (*Ducluzeau et al., 2008*; *de Gier et al., 1996*; *Zufferey et al., 1996*). CcoN is the membrane-integrated catalytic subunit and contains two *b*-type hemes and a copper ion. CcoO is membrane-anchored and contains one *c*-type heme. Additional redox subunits and/or subunits implicated in complex assembly, such as CcoQ and CcoP, can be encoded by adjacent genes (*Figure 1B*). *ccoNO*-containing clusters are widely distributed across phyla of the bacterial domain (*Ducluzeau et al., 2008*). We used the Egg-NOG database, which contains representative genomes for more than 3000 bacterial species (*Huerta-Cepas et al., 2016*) to obtain an overview of the presence and frequency of *cco* genes. Out of 3318 queried bacterial genomes, we found 467 with full *cco* operons (encoding potentially functional $cbb_3$-type oxidases with O and N subunits). Among these, 78 contain more than one full operon. We also used EggNOG to look for orphan *ccoN* genes by examining the relative numbers of *ccoO* and *ccoN* homologs in individual genomes. We found 14 genomes, among which

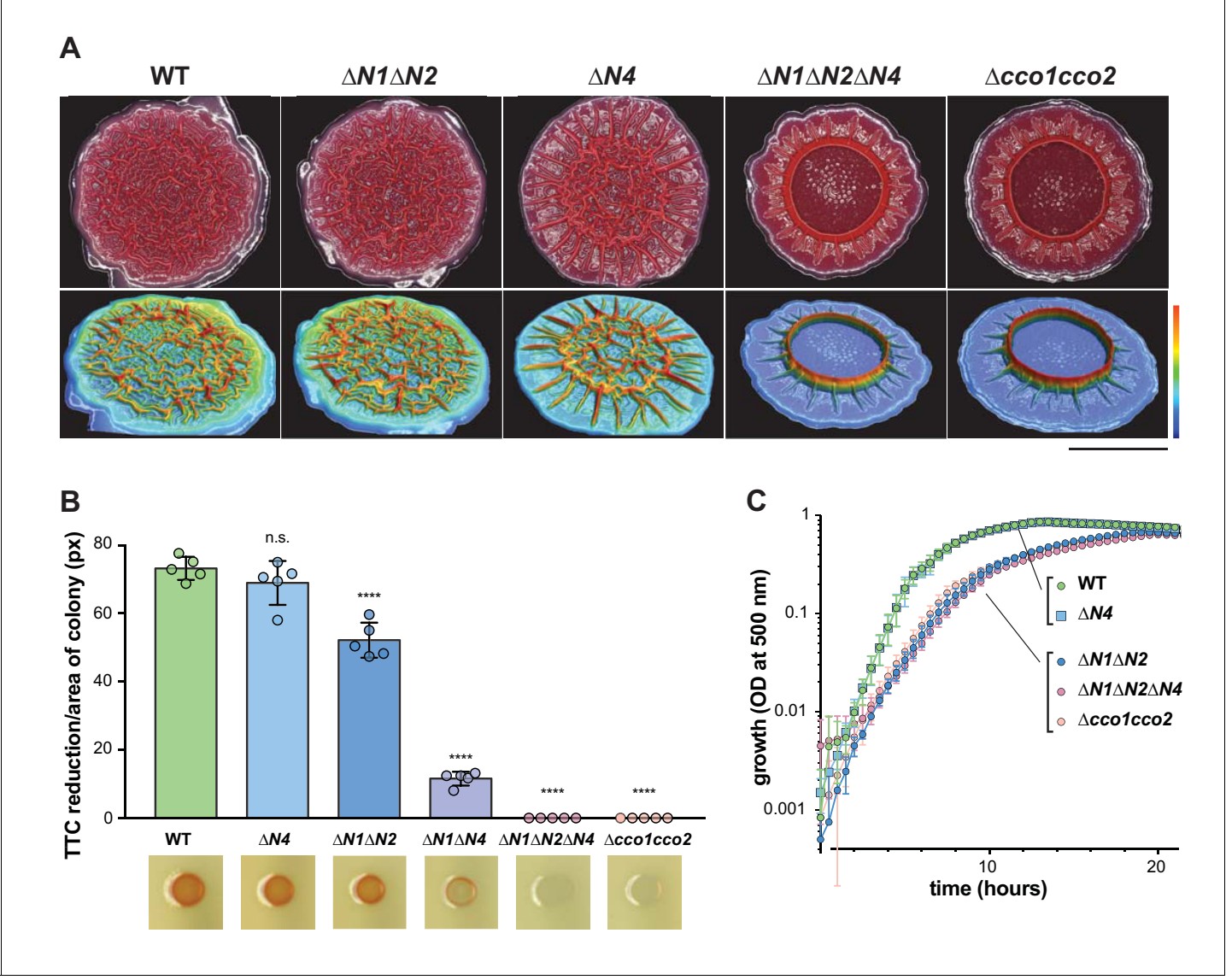

**Figure 2.** CcoN4-containing heterocomplexes make biofilm-specific contributions to morphogenesis and respiration. (A) Top: Five-day-old colony biofilms of PA14 WT and *cco* mutant strains. Biofilm morphologies are representative of more than 10 biological replicates. Images were generated using a digital microscope. Scale bar is 1 cm. Bottom: 3D surface images of the biofilms shown in the top panel. Images were generated using a wide-area 3D measurement system. Height scale bar: bottom (blue) to top (red) is 0–0.7 mm for WT, *ΔN1ΔN2*, and *ΔN4*; 0–1.5 mm for *ΔN1ΔN2ΔN4* and *Δcco1cco2*. (B) TTC reduction by WT and *cco* mutant colonies after 1 day of growth. Upon reduction, TTC undergoes an irreversible color change from colorless to red. Bars represent the average, and error bars represent the standard deviation, of individually-plotted biological replicates (n = 5). p-Values were calculated using unpaired, two-tailed t tests comparing each mutant to WT (****p≤0.0001). (C) Mean growth of PA14 WT and *cco* mutant strains in MOPS defined medium with 20 mM succinate. Error bars represent the standard deviation of biological triplicates.
DOI: https://doi.org/10.7554/eLife.30205.004

The following figure supplements are available for figure 2:

**Figure supplement 1.** Effects of individual and combined *cco* gene deletions on colony biofilm morphogenesis.
DOI: https://doi.org/10.7554/eLife.30205.005

**Figure supplement 2.** PA14 WT, *Δphz*, and *cco* mutant growth phenotypes are unaffected by endogenous cyanide production.
DOI: https://doi.org/10.7554/eLife.30205.006

**Figure supplement 3.** Pseudomonads with CcoN homologs.
DOI: https://doi.org/10.7554/eLife.30205.007

**Figure supplement 4.** Comparison of the PA14 CcoN subunit sequences and analysis of the predicted structure of CcoN4.
DOI: https://doi.org/10.7554/eLife.30205.008

*Pseudomonas* species are overrepresented, that contain orphan *ccoN* genes (*Figure 1C*), and our analysis yielded three species that contain more than one orphan *ccoN* gene: *Pseudomonas mendocina*, *Pseudomonas aeruginosa*, and *Achromobacter xylosoxidans*. *P. mendocina* is a soil bacterium and occasional nosocomial pathogen that is closely related to *P. aeruginosa*, based on 16S rRNA gene sequence comparison (*Anzai et al., 2000*). *A. xylosoxidans*, in contrast, is a member of a different proteobacterial class but nevertheless is often mistaken for *P. aeruginosa* (*Saiman et al., 2001*). Like *P. aeruginosa*, it is an opportunistic pathogen that can cause pulmonary infections in immunocompromised individuals and patients with cystic fibrosis (*De Baets et al., 2007*; *Firmida et al., 2016*). Hirai *et al.* previously reported a ClustalW-based analysis of CcoN homologs specifically from pseudomonads, which indicated the presence of orphan genes in additional species not represented in the EggNOG database. These include *P. denitrificans*, which contains two orphan genes (*Hirai et al., 2016*).

## CcoN4-containing isoforms function specifically in biofilms to support community morphogenesis and respiration

During growth in a biofilm, subpopulations of cells are subjected to regimes of electron donor and $O_2$ availability that may create unique metabolic demands and require modulation of the respiratory chain for survival (*Alvarez-Ortega and Harwood, 2007*; *Borriello et al., 2004*; *Werner et al., 2004*). We therefore investigated the contributions of individual *cco* genes and gene clusters to *P. aeruginosa* PA14 biofilm development using a colony morphology assay, which has demonstrated sensitivity to electron acceptor availability and utilization (*Dietrich et al., 2013*). Because the Cco1 and Cco2 complexes are the most important cytochrome oxidases for growth of *P. aeruginosa* in fully aerated and $O_2$-limited liquid cultures (*Alvarez-Ortega and Harwood, 2007*; *Arai et al., 2014*), we predicted that mutations disabling the functions of Cco1 and Cco2 would affect colony growth. Indeed, a mutant lacking both the *cco1* and *cco2* operons ('Δ*cco1cco2*') produced thin biofilms with a smaller diameter than the wild type. After 5 days of development, this mutant displayed a dramatic phenotype consisting of a tall central ring feature surrounded by short ridges that emanate radially (*Figure 2A*, *Figure 2—figure supplement 1A*). Δ*cco1cco2* colonies were also darker in color, indicating increased uptake of the dye Congo red, which binds to the extracellular matrix produced by biofilms (*Friedman and Kolter, 2004*). Surprisingly, a strain specifically lacking the catalytic subunits of Cco1 and Cco2 ('Δ*N1*Δ*N2*'), while showing a growth defect similar to that of Δ*cco1cco2* when grown in liquid culture (*Figure 2C*), showed biofilm development that was similar to that of the wild type (*Figure 2A*, *Figure 2—figure supplement 1A*).

As it is known that CcoN3 and CcoN4 can form functional complexes with subunits of the Cco1 and Cco2 oxidases in *P. aeruginosa* PAO1 (*Hirai et al., 2016*), this led us to hypothesize that Cco isoforms containing the orphan subunits CcoN3 and/or CcoN4 could substitute for Cco1 and Cco2 in the biofilm context. Deleting *ccoN3* ('Δ*N3*' or 'Δ*N1*Δ*N2*Δ*N3*') did not have an observable effect on biofilm development when mutants were compared to respective parent strains (*Figure 2—figure supplement 1A*). However, the phenotype of a 'Δ*N1*Δ*N2*Δ*N4*' mutant was consistent with our model, as it mimicked that of the Δ*cco1cco2* mutant in both liquid-culture and biofilm growth (*Figure 2A and C*, *Figure 2—figure supplement 1A*). Furthermore, we found that a mutant lacking only *ccoN4* ('Δ*N4*') displayed an altered phenotype in that it began to form wrinkle structures earlier than the wild type (*Figure 2—figure supplement 1A*), which developed into a disordered region of wrinkles inside a central ring, surrounded by long, radially emanating ridges (*Figure 2A*). Reintroduction of the *ccoN4* gene into either of these strains restored the phenotypes of the respective parent strains (*Figure 2—figure supplement 1A*). Deletion of either *ccoN2* or *ccoN3* in the Δ*N4* background did not exacerbate the colony phenotype seen in Δ*N4* alone. However, the 'Δ*N1*Δ*N4*' double mutant showed an intermediate phenotype relative to Δ*N4* and Δ*N1*Δ*N2*Δ*N4* (*Figure 2—figure supplement 1B*), suggesting some functional redundancy for CcoN1 and CcoN4. The developmental pattern of the Δ*N4* colony is reminiscent of those displayed by mutants defective in phenazine production and sensing (*Figure 2—figure supplement 1A*) (*Dietrich et al., 2008*; *2013*; *Sakhtah et al., 2016*; *Okegbe et al., 2017*). Although Δ*N4* itself showed a unique phenotype in the colony morphology assay, its growth in shaken liquid cultures was indistinguishable from that of the wild type (*Figure 2C*). Finally, deleting the three non-*cbb₃*-type terminal oxidases ('Δ*cox*Δ*cyo*Δ*cio*'), did not affect biofilm morphology (*Figure 2—figure supplement 2C*). These results suggest that CcoN4-

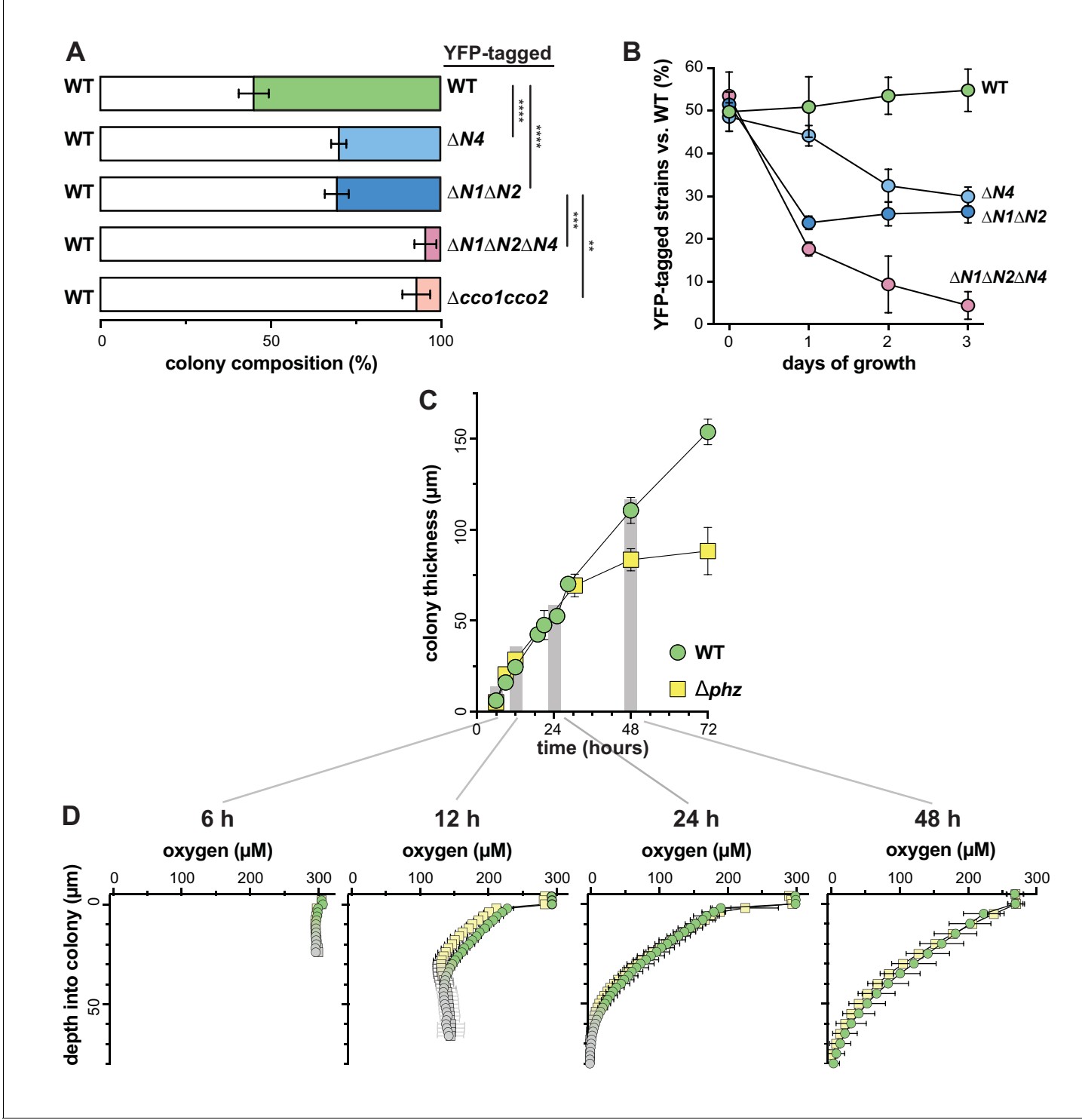

**Figure 3.** CcoN4 confers a competitive advantage in biofilms, particularly when $O_2$ becomes limiting. (**A**) Relative fitness of various YFP-labeled *cco* mutants when co-cultured with WT in mixed-strain biofilms for 3 days. Error bars represent the standard deviation of biological triplicates. p-Values were calculated using unpaired, two-tailed t tests (**$p \leq 0.01$; ***$p \leq 0.001$; ****$p \leq 0.0001$). (**B**) Time course showing relative fitness, over a period of 3 days, of various *cco* mutants when co-cultured with WT in mixed-strain biofilms. Results are shown for experiments in which the WT was co-cultured with various 'labeled' strains, that is, those that were engineered to constitutively express YFP. (See *Figure 3—figure supplement 1* for results from experiments in which the labeled WT was co-cultured with unlabeled mutants.) Error bars represent the standard deviation of biological triplicates. (**C**) Change in thickness over 3 days of development for colony biofilms of WT and Δ*phz* as assessed by thin sectioning and DIC microscopy. After the onset of wrinkling, thickness was determined for the base (i.e. the 'valley' between wrinkles). Error bars represent the standard deviation of biological

*Figure 3 continued on next page*

*Figure 3 continued*

triplicates. (D) O$_2$ profiles of colonies at selected timepoints within the first 3 days of biofilm development. Gray point markers indicate measurements made in the agar directly below the colony. Error bars denote standard deviation of biological triplicates.

DOI: https://doi.org/10.7554/eLife.30205.009

The following figure supplement is available for figure 3:

**Figure supplement 1.** CcoN4 is necessary for optimal fitness in biofilms, particularly when O$_2$ becomes limiting.

DOI: https://doi.org/10.7554/eLife.30205.010

containing Cco isoforms play physiological roles that are specific to the growth conditions encountered in biofilms.

Next, we asked whether CcoN4 contributes to respiration in biofilms. We tested a suite of *cco* mutants for reduction of triphenyl tetrazolium chloride (TTC), an activity that is associated with cytochrome *c* oxidase-dependent respiration (*Rich et al., 2001*). The Δ*cco1cco2* mutant showed a severe defect in TTC reduction, which was recapitulated by the Δ*N1*Δ*N2*Δ*N4* mutant. As in the colony morphology assay, this extreme phenotype was not recapitulated in a mutant lacking only CcoN1 and CcoN2, indicating that CcoN4 contributes to respiratory activity in PA14 biofilms. Although we did not detect a defect in TTC reduction for the Δ*N4* mutant, we saw an intermediate level of TTC reduction for Δ*N1*Δ*N4* compared to Δ*N1*Δ*N2* and Δ*N1*Δ*N2*Δ*N4*, further implicating the CcoN4 subunit in this activity (*Figure 2B*).

A recent study demonstrated a role for CcoN4 in resistance to cyanide, a respiratory toxin that is produced by *P. aeruginosa* (*Hirai et al., 2016*). The altered biofilm phenotypes of Δ*N4* mutants could therefore be attributed to an increased sensitivity to cyanide produced during biofilm growth. We deleted the *hcn* operon, coding for cyanide biosynthetic enzymes, in wild-type, phenazine-null (Δ*phz*), and various *cco* mutant backgrounds. The biofilm morphologies and liquid-culture growth of these strains were unaffected by the Δ*hcnABC* mutation, indicating that the biofilm-specific role of CcoN4 explored in this work is independent of its role in mediating cyanide resistance (*Figure 2—figure supplement 2*). Additionally, we examined genomes available in the Pseudomonas Genome Database for the presence of homologs encoding CcoN subunits (*ccoN* genes) and enzymes for cyanide synthesis (*hcnABC*) (*Winsor et al., 2016*) and did not find a clear correlation between the presence of *hcnABC* and *ccoN4* homologs (*Figure 2—figure supplement 3*).

Together, the effects of *cco* gene mutations that we observed in assays for colony morphogenesis and TTC reduction suggest that one or more CcoN4-containing Cco isoform(s) support respiration and redox balancing, and is/are utilized preferentially in comparison to CcoN1- and CcoN2-containing Cco complexes, in biofilms. We performed a sequence alignment of the CcoN subunits encoded by the PA14 genome and identified residues that are unique to CcoN4 or shared uniquely between CcoN4 and CcoN1, which showed the strongest functional redundancy with CcoN4 in our assays (*Figure 2—figure supplement 4A*). We also threaded the CcoN4 sequence using the available structure of the CcoN subunit from *P. stutzeri* (*Buschmann et al., 2010)* and highlighted these residues (*Figure 2—figure supplement 4B*). It is noteworthy that most of the highlighted residues are surface-exposed, specifically on one half of the predicted CcoN4 structure, where they may engage in binding an unknown protein partner or specific lipids. In contrast, sites that have been described as points of interaction with CcoO and CcoP are mostly conserved, further supporting the notion that CcoN4 can interact with these subunits in Cco complexes.

## Different CcoN subunits are required for competitive fitness in early or late colony development

To further test CcoN4's contribution to growth in biofilms, we performed competition assays in which Δ*N4* and other mutants were grown as mixed-strain biofilms with the wild type. In each of these assays, one strain was labeled with constitutively expressed YFP so that the strains could be distinguished during enumeration of colony-forming units (CFUs). Experiments were performed with the label on each strain to confirm that YFP expression did not affect fitness (*Figure 3—figure supplement 1A,B*). When competitive fitness was assessed after 3 days of colony growth (*Figure 3A*), Δ*N4* cells showed a disadvantage, with the wild type outcompeting Δ*N4* by a factor of two. This was similar to the disadvantage observed for the Δ*N1*Δ*N2* mutant, further suggesting that the orphan

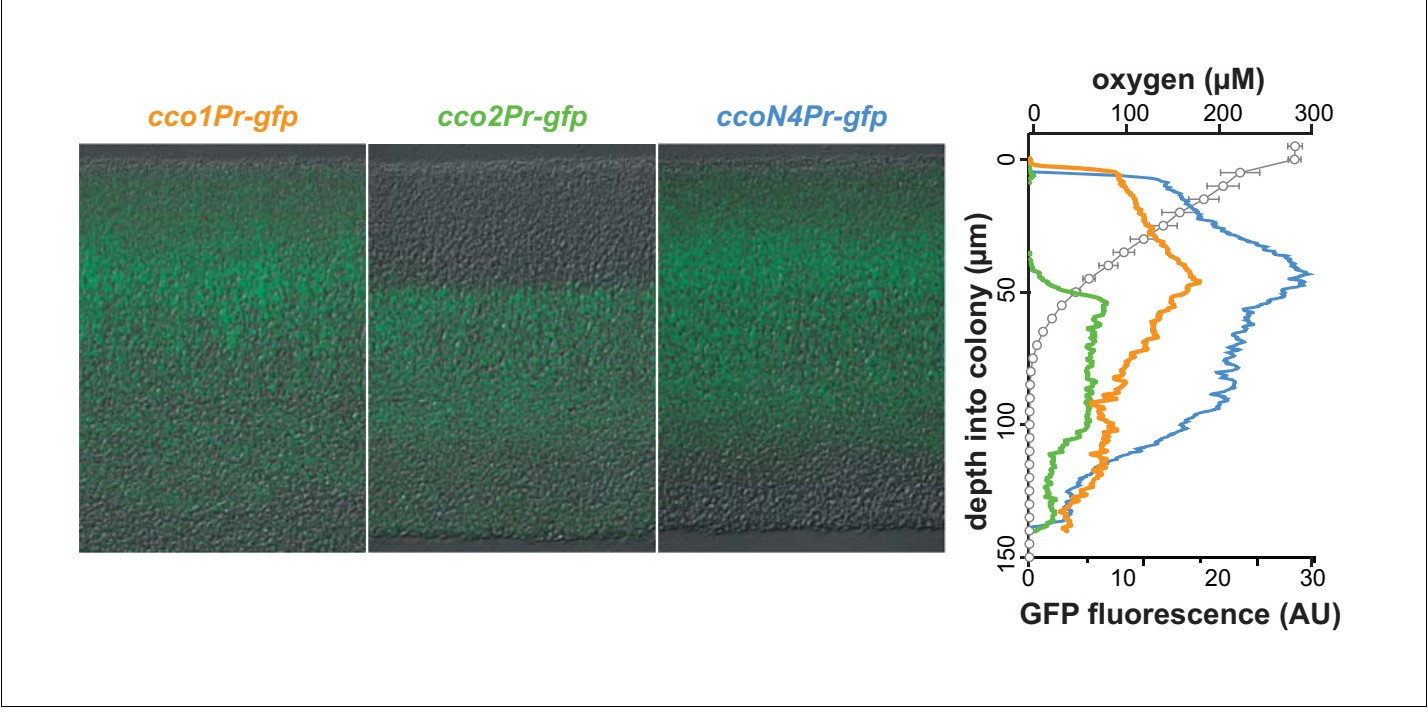

**Figure 4.** *cco* genes are differentially expressed over biofilm depth. Left: Representative images of thin sections prepared from WT biofilms grown for 3 days. Each biofilm is expressing a translational GFP reporter under the control of the *cco1*, *cco2*, or *ccoN4Q4* promoter. Reporter fluorescence is shown in green and overlain on respective DIC images. Right: Fluorescence values corresponding to images on the left. Fluorescence values for a strain containing the *gfp* gene without a promoter (the empty MCS control) have been subtracted from each respective plot. $O_2$ concentration over depth (open circles) from 3-day-old WT biofilms is also shown. Error bars represent the standard deviation of biological triplicates and are not shown in cases where they would be obscured by the point markers. y-axis in the right panel provides a scale bar for the left panel. Reporter fluorescence images and values are representative of 4 biological replicates.

DOI: https://doi.org/10.7554/eLife.30205.011

The following figure supplement is available for figure 4:

**Figure supplement 1.** Expression of *cco* reporters in shaken liquid cultures.

DOI: https://doi.org/10.7554/eLife.30205.012

subunit CcoN4 plays a significant role in biofilm metabolism. Remarkably, deletion of *ccoN4* in mutants already lacking *ccoN1* and *ccoN2* led to a drastic decrease in fitness, with the wild type out-competing Δ*N1*Δ*N2*Δ*N4* by a factor of 16. This disadvantage was comparable to that observed for the mutant lacking the full *cco* operons (Δ*cco1cco2*), underscoring the importance of CcoN4-containing isoforms during biofilm growth.

To further explore the temporal dynamics of N subunit utilization, we repeated the competition assay, but sampled each day over the course of 3 days (**Figure 3B**). The fitness disadvantage that we had found for strains lacking CcoN1 and CcoN2 was evident after only 1 day of growth and did not significantly change after that. In contrast, the Δ*N4*-specific decline in fitness did not occur before the second day. These data suggest that the contributions of the various N subunits to biofilm metabolism differ depending on developmental stage.

DIC imaging of thin sections from wild-type colonies reveals morphological variation over depth that may result from decreasing $O_2$ availability (**Figure 3—figure supplement 1C**). We have previously reported that 3-day-old PA14 colony biofilms are hypoxic at depth (**Dietrich et al., 2013**) and that $O_2$ availability is generally higher in thinner biofilms, such as those formed by the phenazine-null mutant Δ*phz*. We have proposed that the utilization of phenazines as electron acceptors in wild-type biofilms enables cellular survival in the hypoxic zone and promotes colony growth (**Okegbe et al., 2014**). The relatively late-onset phenotype of the Δ*N4* mutant in the competition assay suggested to us that CcoN4 may play a role in survival during formation of the hypoxic colony subzone and that this zone could arise at a point between 1 and 2 days of colony growth. We measured $O_2$

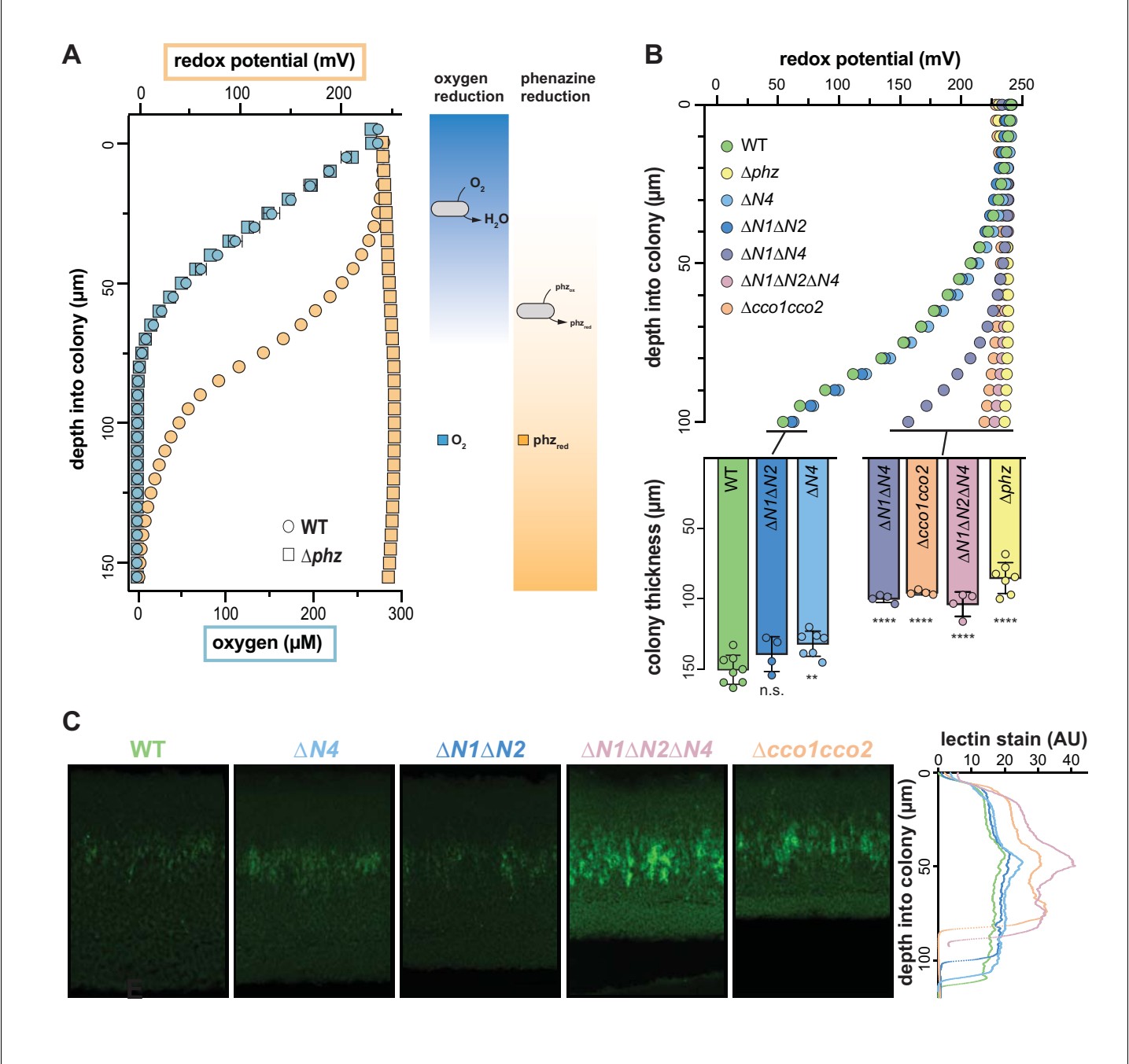

**Figure 5.** Characterization of chemical gradients and matrix distribution in PA14 WT and mutant colony biofilms. (**A**) Left: Change in $O_2$ concentration (blue) and redox potential (orange) with depth for WT and $\Delta phz$ biofilms grown for two days. WT biofilms are ~150 μm thick while $\Delta phz$ biofilms are ~80 μm thick. For $O_2$ profiles, error bars represent the standard deviation of biological triplicates. For redox profiles, data are representative of at least 5 biological replicates. Right: model depicting the distribution of $O_2$ and reduced vs. oxidized phenazines in biofilms. (**B**) Top: Change in redox potential with depth for WT and various mutant biofilms grown for 2 days. Data are representative of at least 5 biological replicates. Bottom: Thickness of 3-day-old colony biofilms of the indicated strains. Bars represent the average of the plotted data points (each point representing a biological replicate, n ≥ 4), and error bars represent the standard deviation. p-Values were calculated using unpaired, two-tailed t tests comparing each mutant to WT (n.s., not significant; **p≤0.01; ****p≤0.0001). (**C**) Left: Representative thin sections of WT and *cco* mutant biofilms, stained with lectin and imaged by fluorescence microscopy. Biofilms were grown for 2 days before sampling. Right: Relative quantification of lectin stain signal intensity. Coloration of strain names in the left panel provides a key for the plotted data, and the y-axis in the right panel provides a scale bar for the left panel. Lectin-staining images and values are representative of 4 biological replicates.

DOI: https://doi.org/10.7554/eLife.30205.013

*Figure 5 continued on next page*

*Figure 5 continued*

The following figure supplement is available for figure 5:

**Figure supplement 1.** Use of a redox microelectrode to measure phenazine reduction in colony biofilms.
DOI: https://doi.org/10.7554/eLife.30205.014

concentrations in wild-type and Δ*phz* biofilms at specific time points over development, and found that $O_2$ declined similarly with depth in both strains (*Figure 3D*). The rate of increase in height of Δ*phz* tapered off when a hypoxic zone began to form, consistent with the notion that the base does not increase in thickness when electron acceptors ($O_2$ or phenazines) are not available. Although we cannot pinpoint the exact depth at which the $O_2$ microsensor leaves the colony base and enters the underlying agar, we can estimate these values based on colony thickness measurements (*Figure 3C*). When we measured the thickness of wild-type and Δ*phz* biofilms over 3 days of incubation, we found that the values began to diverge between 30 and 48 hr of growth, after the colonies reached ~70 μm in height, which coincides with the depth at which $O_2$ becomes undetectable. Δ*phz* colonies reached a maximum thickness of ~80 μm, while wild-type colonies continued to grow to ~150 μm (*Figure 3C*). In this context, it is interesting to note that the point of divergence for the increase in wild-type and Δ*phz* colony thickness—between 30 and 48 hr—corresponds to the point at which CcoN4 becomes important for cell viability in our mixed-strain colony growth experiments (*Figure 3B*). We hypothesize that this threshold thickness leads to a level of $O_2$ limitation that is physiologically relevant for the roles of phenazines and CcoN4 in biofilm metabolism.

## *cco* genes show differential expression across biofilm subzones

*P. aeruginosa*'s five canonical terminal oxidases are optimized to function under and in response to distinct environmental conditions, including various levels of $O_2$ availability (*Arai et al., 2014*; *Kawakami et al., 2010*; *Alvarez-Ortega and Harwood, 2007*; *Comolli and Donohue, 2004*). Furthermore, recent studies, along with our results, suggest that even within the Cco terminal oxidase complexes, the various N subunits may perform different functions (*Hirai et al., 2016*). We sought to determine whether differential regulation of *cco* genes could lead to uneven expression across biofilm subzones. To test this, we engineered reporter strains in which GFP expression is regulated by the *cco1*, *cco2*, or *ccoN4Q4* promoters. Biofilms of these strains were grown for 3 days, thin-sectioned, and imaged by fluorescence microscopy. Representative results are shown in the left panel of *Figure 4*. The right panel of *Figure 4* contains plotted GFP signal intensity and $O_2$ concentration measurements over depth for PA14 wild-type colonies. *cco1* and *ccoN4* expression patterns indicate that the Cco1 oxidase and the CcoN4 subunit are produced throughout the biofilm (*Figure 4*). *cco2* expression, on the other hand, is relatively low in the top portion of the biofilm and shows a sharp induction starting at a depth of ~45 μm. This observation is consistent with previous studies showing that *cco2* expression is regulated by Anr, a global transcription factor that controls gene expression in response to a shift from oxic to anoxic conditions (*Comolli and Donohue, 2004*; *Kawakami et al., 2010*; *Ray and Williams, 1997*).

Although previous studies have evaluated expression as a function of growth phase in shaken liquid cultures for *cco1* and *cco2*, this property has not been examined for *ccoN4Q4*. We monitored the fluorescence of our engineered *cco* gene reporter strains during growth under this condition in a nutrient-rich medium. As expected based on the known constitutive expression of *cco1* and Anr-dependence of *cco2* induction, we saw *cco1*-associated fluorescence increase before that associated with *cco2*. Induction of *ccoN4Q4* occurred after that of *cco1* and *cco2* (*Figure 4—figure supplement 1*), consistent with microarray data showing that this locus is strongly induced by $O_2$ limitation (*Alvarez-Ortega and Harwood, 2007*). However, our observation that *ccoN4Q4* is expressed in the aerobic zone, where *cco2* is not expressed, in biofilms (*Figure 4*) suggests that an Anr-independent mechanism functions to induce this operon during multicellular growth.

Our results indicate that different Cco isoforms may function in specific biofilm subzones, but that CcoN4-containing isoforms could potentially form throughout the biofilm. These data, together with our observation that Δ*N4* biofilms exhibit a fitness disadvantage from day 2 (*Figure 3B*), led us to more closely examine the development and chemical characteristics of the biofilm over depth.

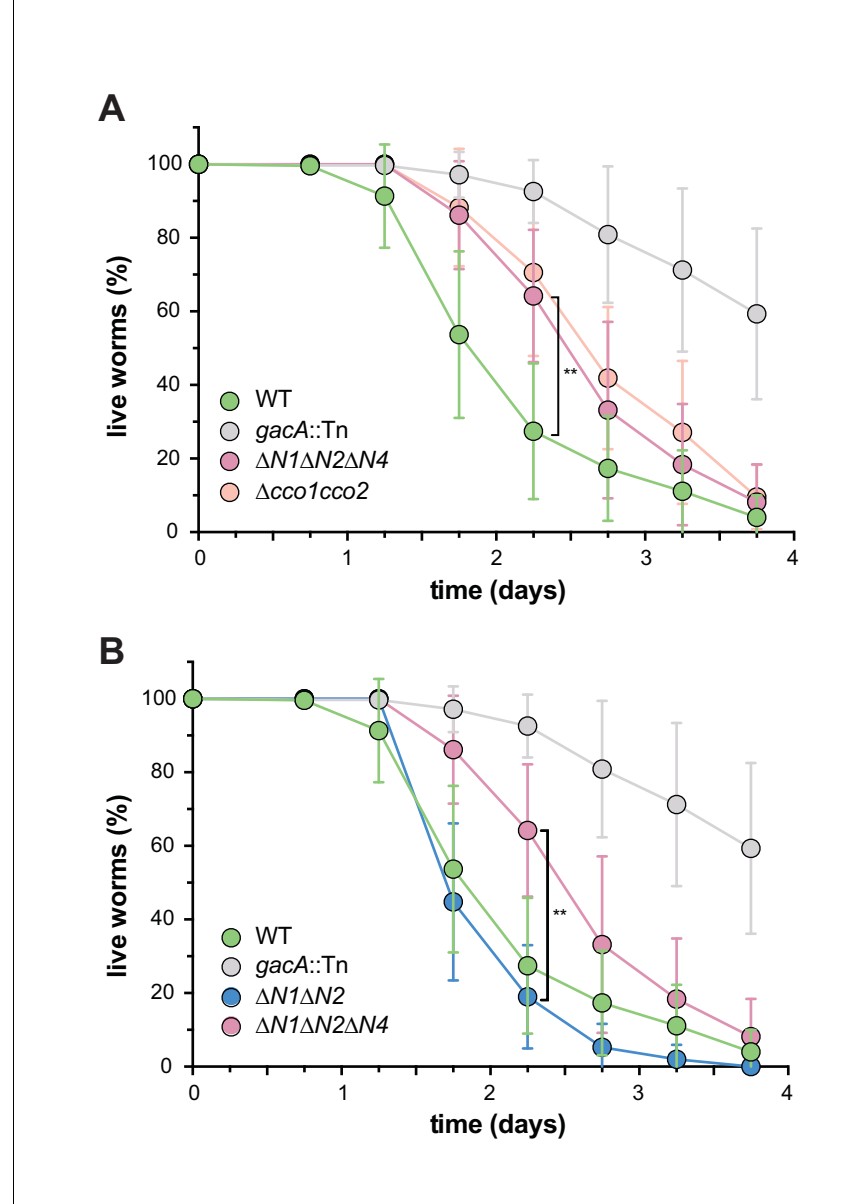

**Figure 6.** CcoN4-containing isoform(s) make unique contributions to PA14 virulence. Slow-killing kinetics of WT, *gacA*, and various *cco* mutant strains in the nematode *Caenorhabditis elegans*. Nearly 100% of the *C. elegans* population exposed to WT PA14 is killed after 4 days of exposure to the bacterium, while a mutant lacking GacA, a regulator that controls expression of virulence genes in *P. aeruginosa*, shows decreased killing, with ~50% of worms alive 4 days post-exposure. (A) ΔN1ΔN2ΔN4 and Δcco1cco2 show comparably attenuated pathogenicity relative to WT. Error bars represent the standard deviation of at least 6 biological replicates. At 2.25 days post-exposure, significantly less *C. elegans* were killed by ΔN1ΔN2ΔN4 than by WT (unpaired two-tailed t test; p=0.0022). (B) ΔN1ΔN2 displays only slightly reduced pathogenicity when compared to WT. At 2.25 days post-exposure, significantly more *C. elegans* were killed by ΔN1ΔN2 than by ΔN1ΔN2ΔN4 (unpaired two-tailed t test; p=0.003). Error bars represent the standard deviation of at least 4 biological replicates, each with a starting sample size of 30–35 worms per replicate.
DOI: https://doi.org/10.7554/eLife.30205.015

## Microelectrode-based redox profiling reveals differential phenazine reduction activity in wild-type and *cco* mutant biofilms

The results shown in *Figure 2B* implicate CcoN4-containing isoforms in the reduction of TTC, a small molecule that interacts with the respiratory chain (*Rich et al., 2001*). Similar activities have been demonstrated for phenazines, including the synthetic compound phenazine methosulfate (PMS) (*Nachlas et al., 1960*) and those produced naturally by *P. aeruginosa* (*Armstrong and Stewart-Tull, 1971*). Given that CcoN4 and phenazines function to influence morphogenesis at similar stages of biofilm growth (*Figures 2A* and *3*, *Figure 2—figure supplement 1*, *Figure 3—figure supplement 1A,B*), we wondered whether the role of CcoN4 in biofilm development was linked to phenazine metabolism. We used a Unisense platinum microelectrode with a 20–30 µm tip to measure the extracellular redox potential in biofilms as a function of depth. This electrode measures the inclination of the sample to donate or accept electrons relative to a Ag/AgCl reference electrode. We found that wild-type colonies showed a decrease in redox potential over depth, indicating an increased ratio of reduced to oxidized phenazines, while the redox potential of Δ*phz* colonies remained unchanged (*Figure 5A*). To confirm that phenazines are the primary determinant of the measured redox potential in the wild type, we grew Δ*phz* colonies on medium containing PMS (which resembles the natural phenazines that regulate *P. aeruginosa* colony morphogenesis [*Sakhtah et al., 2016*]) and found that these colonies yielded redox profiles similar to those of the wild type (*Figure 5—figure supplement 1A*). Therefore, although the microelectrode we employed is capable of interacting with many redox-active substrates, we found that its signal was primarily determined by phenazines in our system. In addition, while wild-type colonies showed rapid decreases in $O_2$ availability starting at the surface, the strongest decrease in redox potential was detected after ~50 µm (*Figure 5A*). These results suggest that the bacteria residing in the biofilm differentially utilize $O_2$ and phenazines depending on their position and that $O_2$ is the preferred electron acceptor.

We hypothesized that one or more of the CcoN subunits encoded by the PA14 genome is required for phenazine reduction and tested this by measuring the redox potential over depth for a series of *cco* mutants (*Figure 5B*, top). We saw very little reduction of phenazines in the Δ*cco1cco2* colony, suggesting that $cbb_3$-type oxidases are required for this activity. In contrast, the mutant lacking the catalytic subunits of Cco1 and Cco2, Δ*N1*Δ*N2*, showed a redox profile similar to the wild type, indicating that isoforms containing one or both of the orphan CcoN subunits could support phenazine reduction activity. Indeed, although redox profiles obtained for the Δ*N1*Δ*N2* and Δ*N4* mutants were similar to those obtained for the wild type, the redox profile of the Δ*N1*Δ*N2*Δ*N4* mutant recapitulated that of Δ*cco1cco2*. These results indicate redundancy in the roles of some of the CcoN subunits. Consistent with this, Δ*N1*Δ*N4* showed an intermediate defect in phenazine reduction. We note that the triple mutant Δ*cox*Δ*cyo*Δ*cio* showed a wild-type-like redox profile, indicating that the $cbb_3$-type terminal oxidases are sufficient for normal phenazine reduction (*Figure 5—figure supplement 1B*). Extraction and measurement of phenazines released from wild-type and *cco* mutant biofilms showed that variations in redox profiles could not be attributed to differences in phenazine production (*Figure 5—figure supplement 1C*).

Our group has previously shown that a Δ*phz* mutant compensates for its lack of phenazines by forming thinner colonies, thus limiting the development of the hypoxic subzone seen in the wild type (*Dietrich et al., 2013*). We therefore hypothesized that mutants unable to reduce phenazines would likewise result in thinner colonies. Indeed, we observed that the *cco* mutants that lacked phenazine reduction profiles in the top panel of *Figure 5B* produced biofilms that were significantly thinner than wild-type and comparable to that of the Δ*phz* mutant (*Figure 5B*, bottom).

Our group has also reported that reduction of nitrate, an alternate electron acceptor for *P. aeruginosa* (*Williams et al., 2007*), can serve as an additional redox-balancing strategy for cells in biofilms (*Dietrich et al., 2013*). Colony wrinkling is stimulated by a reduced cellular redox state; thus, provision of nitrate in the growth medium inhibits colony feature formation. We hypothesized that nitrate reduction could compensate for defects in $O_2$ and phenazine reduction and inhibit colony wrinkling in the *cco* mutants that are the focus of this study. To test this, we grew strains on medium containing 10 or 40 mM potassium nitrate. We found that 10 mM nitrate was sufficient to inhibit wrinkling for up to 4 days of incubation in the wild type, Δ*N4*, and Δ*N1*Δ*N4*, but that Δ*phz* and Δ*N1*Δ*N2*Δ*N4* had initiated wrinkling at this point (*Figure 5—figure supplement 1D*). When we grew these strains on medium containing 40 mM nitrate, we saw increased inhibition of wrinkling such

that the wild type, $\Delta phz$, $\Delta N4$, and $\Delta N1\Delta N4$ remained completely smooth at 4 days of incubation. Although $\Delta N1\Delta N2\Delta N4$ had shown some feature formation after 4 days on this medium, it was diminished relative to the same point on 10 mM nitrate. These results suggest that $O_2$ reduction, phenazine reduction, and nitrate reduction can operate in synchrony to oxidize the redox states of cells in biofilms and that provision of nitrate can compensate for defects in $O_2$ and phenazine reduction to enable maintenance of redox homeostasis.

## Wild-type and *cco* mutant colony biofilms show increased matrix production at comparable depths

We have recently demonstrated that extracellular matrix production, a hallmark of biofilm formation, is regulated by redox state in PA14 colony biofilms. Increased matrix production correlates with the accumulation of reducing power (as indicated by higher cellular $NADH/NAD^+$ ratios) due to electron acceptor limitation and is visible in the hypoxic region of $\Delta phz$ colonies (*Dietrich et al., 2013*; *Okegbe et al., 2017*). The morphologies of our *cco* mutants (*Figure 2A*) suggest that matrix production can also be induced by respiratory chain dysfunction, which may be linked to defects in phenazine utilization (*Figure 5B*). To further examine the relationships between Cco isoforms and redox imbalance in biofilms, we prepared thin sections from 2-day-old colonies and stained with fluorescein-labeled lectin, which binds preferentially to the Pel polysaccharide component of the matrix (*Jennings et al., 2015*). Consistent with their similar gross morphologies, the wild-type and $\Delta N1\Delta N2$ biofilms showed similar patterns of staining, with a faint band of higher intensity at a depth of ~40 µm (*Figure 5C*). $\Delta N4$ also showed a similar pattern, with a slightly higher intensity of staining in this band. $\Delta N1\Delta N2\Delta N4$ and $\Delta cco1cco2$ showed more staining throughout each sample, with wider bands of greater intensity at the ~40 µm point. These data suggest that deletion of the Cco complexes leads to a more reduced cellular redox state, which induces production of more matrix, and that CcoN4 contributes significantly to maintaining redox homeostasis when $O_2$ is limiting.

## CcoN4 contributes to *P. aeruginosa* virulence in a *C. elegans* slow killing model

We have previously shown that a mutant defective in biofilm-specific phenazine production, which also shows altered colony morphology (*Dietrich et al., 2008*; *2013*), exhibits decreased virulence (*Recinos et al., 2012*). We and others have suggested that one way in which phenazines could contribute to virulence is by acting as electron acceptors to balance the intracellular redox state in the hypoxic conditions that are encountered during infection (*Price-Whelan et al., 2006*; *Newman, 2008*; *Dietrich et al., 2013*). Because CcoN4 is required for wild-type biofilm architecture and respiration (*Figures 2A, C* and *5C*), we hypothesized that it could also contribute to virulence. To test this, we conducted virulence assays using the nematode *Caenorhabditis elegans* as a host. It has been shown that *P. aeruginosa* is pathogenic to *C. elegans* and that the slow killing assay mimics an infection-like killing of *C. elegans* by the bacterium (*Tan et al., 1999*). While $\Delta N1\Delta N2$ killed with wild-type-like kinetics, $\Delta N1\Delta N2\Delta N4$ and $\Delta cco1cco2$ showed comparably-impaired killing relative to wild-type PA14 (*Figure 6*).

## Discussion

Biofilm formation contributes to *P. aeruginosa* pathogenicity and persistence during different types of infections, including the chronic lung colonizations seen in individuals with cystic fibrosis (*Tolker-Nielsen, 2014*; *Rybtke et al., 2015*). The conditions found within biofilm microenvironments are distinct from those in well-mixed liquid cultures with respect to availability of electron donors and acceptors. We have previously described the roles of phenazines, electron-shuttling antibiotics produced by *P. aeruginosa*, in biofilm-specific metabolism. In this study, we focused on *P. aeruginosa*'s large complement of genes encoding $cbb_3$-type cytochrome oxidase subunits and set out to test their contributions to metabolic electron flow in biofilms.

The *P. aeruginosa* genome contains four different homologs of *ccoN*, encoding the catalytic subunit of $cbb_3$-type oxidase. Only two of these (*ccoN1* and *ccoN2*) are co-transcribed with a *ccoO* homolog, encoding the other critical component of an active $cbb_3$-type oxidase (*Figure 1B*). However, genetic studies have demonstrated that all four versions of CcoN can form functional complexes when expressed with either of the two CcoO homologs (*Hirai et al., 2016*). In well-mixed

liquid cultures, mutants lacking the 'orphan' subunits did not show growth defects (*Figure 2C*) (*Hirai et al., 2016*). We were therefore surprised to find that the $\Delta N4$ mutant showed a unique morphotype in a colony biofilm assay (*Figure 2A*, *Figure 2—figure supplement 1A*). We have applied this assay extensively in our studies of the mechanisms underlying cellular redox balancing and sensing and noted that the phenotype of $\Delta N4$ was similar to that of mutants with defects in electron shuttling and redox signaling (*Dietrich et al., 2013*; *Okegbe et al., 2017*).

We characterized the effects of a $\Delta N4$ mutation on biofilm physiology through a series of assays. In well-mixed liquid cultures, $\Delta cco1cco2$ showed a growth phenotype similar to that of $\Delta N1\Delta N2$. While Hirai et al. have shown that wild-type *P. aeruginosa* cultures grown planktonically do form Cco heterocomplexes containing CcoN4, our observations suggest that such complexes do not contribute significantly to growth under these conditions. Consistent with this, deleting *ccoN4* in the $\Delta N1\Delta N2$ background had no effect on planktonic growth (*Figure 2C*). However, in biofilm-based experiments, we found that deleting *N4* alone was sufficient to cause an altered morphology phenotype (*Figure 2A* and *Figure 2—figure supplement 1A*), and that deleting *N4* in either a $\Delta N1$ or a $\Delta N1\Delta N2$ background profoundly affected biofilm physiology. These experiments included quantification of respiratory activity in colonies, in which deletion of CcoN4 led to a significant decrease (*Figure 2B*); biofilm co-culturing, in which CcoN4 was required for competitive fitness (*Figure 3A and B*, *Figure 3—figure supplement 1*); redox profiling, which showed that CcoN4 can contribute to phenazine reduction (*Figure 5B*, top); colony thickness measurements, which showed that CcoN4 is required for the formation of the hypoxic and anoxic zones (*Figure 5B*, bottom); and matrix profiling, which showed that CcoN4 contributes to the repression of Pel polysaccharide production (*Figure 5C*). The overlap in zones of expression between *cco1*, *cco2*, and *ccoN4Q4* seen in colony thin sections (*Figure 4*) implies that CcoN4 can form heterocomplexes with Cco1 and Cco2 subunits that span the depth of the colony and function to influence the physiology of *P. aeruginosa* biofilms in these ways.

The mutant phenotypes and gene expression profiles reported in this study suggest roles for CcoN4 in $O_2$ and phenazine reduction specifically in the biofilm context, and allow us to draw conclusions about the roles of other CcoN subunits. The expression of *ccoN4Q4* throughout the biofilm depth suggests that CcoN4-containing isoforms could contribute to cytochrome *c* oxidation in both oxic and hypoxic zones (*Figure 4*). This constitutes a deviation from the previously published observation that these genes are specifically induced in hypoxic liquid cultures when compared to well-aerated ones (*Alvarez-Ortega and Harwood, 2007*). Therefore, the *ccoN4Q4* expression we observed in the relatively oxic, upper portion of the colony may be specific to biofilms.

$\Delta N4$ displayed a colony morphology indicative of redox stress and had a fitness disadvantage compared to the wild type (*Figures 2A* and *3A,B*, *Figure 5B*, bottom, *Figure 3—figure supplement 1*). However, because it did not show a defect in phenazine reduction (*Figure 5B*, top), we attribute its colony morphology and impaired fitness phenotypes to its proposed role in $O_2$ reduction (*Hirai et al., 2016*). Similarly, $\Delta N1\Delta N2$ showed reduced fitness compared to the wild type (*Figure 3A and B*, *Figure 3—figure supplement 1*) while showing phenazine reduction comparable to that of the wild type (*Figure 5B*), implying that one or both of these subunits contribute to $O_2$ reduction in biofilms. When CcoN4 was deleted in conjunction with CcoN1 and CcoN2, however, the resulting strain showed a severe phenazine reduction defect, a phenotype recapitulated by deleting both *cco* operons (*Figure 5B*). Thus, our observations suggest a role for the *cbb3*-type oxidases in phenazine reduction in addition to their established roles in $O_2$ reduction, thereby expanding our understanding of their overall contributions *P. aeruginosa*'s physiology and viability.

The results described here can inform our model of how cells survive under distinct conditions in the microenvironments within biofilms. Previous work has shown that pyruvate fermentation can support survival of *P. aeruginosa* under anoxic conditions (*Eschbach et al., 2004*) and that phenazines facilitate this process (*Glasser et al., 2014*). Additional research suggests that phenazine reduction is catalyzed adventitiously by *P. aeruginosa* flavoproteins and dehydrogenases (*Glasser et al., 2017*). Our observation that *cbb3*-type cytochrome oxidases, particularly those containing the CcoN1 or CcoN4 subunits, were required for phenazine reduction in hypoxic biofilm subzones (*Figure 5B*) further implicates the electron transport chain in utilization of these compounds. It is also interesting in light of the historical roles of phenazines acting as mediators in biochemical studies of the cytochrome *bc1* complex and cytochrome oxidases (*King, 1963*; *Armstrong and Stewart-Tull, 1971*; *Davidson et al., 1992*). Based on this earlier work, we can speculate that different CcoN

subunits may indirectly influence phenazine reduction, which could occur at the cytochrome $c$ binding site of the CcoO subunit or elsewhere in the electron transport chain, through effects these CcoN subunits have on the overall function or stability of respiratory complexes. Ultimately, various mechanisms of phenazine reduction and phenazine-related metabolisms may be relevant at different biofilm depths or depending on electron donor availability. Our results suggest that, in the colony biofilm system, enzyme complexes traditionally considered to be specific to $O_2$ reduction may contribute to anaerobic survival.

Because biofilm formation is often associated with colonization of and persistence in hosts, we tested whether CcoN4 contributes to *P. aeruginosa* pathogenicity in *C. elegans*. Similar to our observations in biofilm assays, we found that the Δ*cco1cco2* mutant displayed a more severe phenotype than the Δ*N1*Δ*N2* mutant, suggesting that an orphan subunit can substitute for those encoded by the *cco1* and *cco2* operons. We also found that deleting *ccoN4* in Δ*N1*Δ*N2* led to a Δ*cco1cco2*-like phenotype, suggesting that CcoN4 is the subunit that can play this role (*Figure 6*). In host microenvironments where $O_2$ is available, CcoN4-containing isoforms could contribute to its reduction. Additionally, in hypoxic zones, CcoN4-containing isoforms could facilitate the reduction of phenazines, enabling cellular redox balancing. Both these functions would contribute to persistence of the bacterium within the host. The contributions of the $cbb_3$-type oxidases to *P. aeruginosa* pathogenicity raise the possibility that compounds interfering with Cco enzyme function could be effective therapies for these infections. Such drugs would be attractive candidates due to their specificity for bacterial respiratory chains and, as such, would not affect the host's endogenous respiratory enzymes.

Our discovery that an orphan $cbb_3$-type oxidase subunit contributes to growth in biofilms further expands the scope of *P. aeruginosa*'s remarkable respiratory flexibility. Beyond modularity at the level of the terminal enzyme complex (e.g. utilization of an $aa_3$- vs. a $cbb_3$-type oxidase), the activity of *P. aeruginosa*'s respiratory chain is further influenced by substitution of orphan $cbb_3$-type catalytic subunits for native ones. Utilization of CcoN4-containing isoforms promotes phenazine reduction activity and may influence aerobic respiration in *P. aeruginosa* biofilms. For the exceptional species that contain orphan $cbb_3$-type catalytic subunits, this fine level of control could be particularly advantageous during growth and survival in environments covering a wide range of electron acceptor availability (*Cowley et al., 2015*).

# Materials and methods

**Key resource table**

| Reagent type (species) or resource | Designation | Source or reference | Identifiers | Additional information |
|---|---|---|---|---|
| strain, strain background (UCBPP-PA14 *Pseudomonas aeruginosa*) | wild type (WT) | PMID: 7604262 | | |
| strain, strain background (UCBPP-PA14 *P. aeruginosa*) | Δ*phz* | PMID: 16879411 | | LD24 |
| strain, strain background (UCBPP-PA14 *P. aeruginosa*) | Δ*ccoN1*; Δ*N1* | this study | | LD1784 |
| strain, strain background (UCBPP-PA14 *P. aeruginosa*) | Δ*ccoN2*; Δ*N2* | this study | | LD1614 |
| strain, strain background (UCBPP-PA14 *P. aeruginosa*) | Δ*ccoN3*; Δ*N3* | this study | | LD1620 |
| strain, strain background (UCBPP-PA14 *P. aeruginosa*) | Δ*ccoN4*; Δ*N4* | this study | | LD2833 |
| strain, strain background (UCBPP-PA14 *P. aeruginosa*) | Δ*ccoN1*Δ*ccoN2*; Δ*N1*Δ*N2* | this study | | LD1888 |
| strain, strain background (UCBPP-PA14 *P. aeruginosa*) | Δ*ccoN1*Δ*ccoN4*; Δ*N1*Δ*N4* | this study | | LD1951 |
| strain, strain background (UCBPP-PA14 *P. aeruginosa*) | Δ*ccoN2*Δ*ccoN4*; Δ*N2*Δ*N4* | this study | | LD1692 |

*Continued on next page*

*Continued*

| Reagent type (species) or resource | Designation | Source or reference | Identifiers | Additional information |
|---|---|---|---|---|
| strain, strain background (UCBPP-PA14 *P. aeruginosa*) | ΔccoN3ΔccoN4; ΔN3ΔN4 | this study | | LD1649 |
| strain, strain background (UCBPP-PA14 *P. aeruginosa*) | ΔccoN1ΔccoN2ΔccoN3; ΔN1ΔN2ΔN3 | this study | | LD1977 |
| strain, strain background (UCBPP-PA14 *P. aeruginosa*) | ΔccoN1ΔccoN2ΔccoN4; ΔN1ΔN2ΔN4 | this study | | LD1976 |
| strain, strain background (UCBPP-PA14 *P. aeruginosa*) | ΔccoN1ΔccoN2ΔccoN4 ΔccoN3; ΔN1ΔN2ΔN4ΔN3 | this study | | LD2020 |
| strain, strain background (UCBPP-PA14 *P. aeruginosa*) | Δcco1cco2 | this study | | LD1933 |
| strain, strain background (UCBPP-PA14 *P. aeruginosa*) | ΔcoxΔcyoΔcio | this study | | LD2587 |
| strain, strain background (UCBPP-PA14 *P. aeruginosa*) | Δhcn | this study | | LD2827 |
| strain, strain background (UCBPP-PA14 *P. aeruginosa*) | ΔphzΔhcn | this study | | LD2828 |
| strain, strain background (UCBPP-PA14 *P. aeruginosa*) | ΔccoN4Δhcn; ΔN4Δhcn | this study | | LD2829 |
| strain, strain background (UCBPP-PA14 *P. aeruginosa*) | ΔccoN1ΔccoN2Δhcn; ΔN1ΔN2Δhcn | this study | | LD2830 |
| strain, strain background (UCBPP-PA14 *P. aeruginosa*) | ΔccoN1ΔccoN2ΔccoN4 Δhcn; ΔN1ΔN2ΔN4Δhcn | this study | | LD2831 |
| strain, strain background (UCBPP-PA14 *P. aeruginosa*) | Δcco1cco2Δhcn | this study | | LD2832 |
| strain, strain background (UCBPP-PA14 *P. aeruginosa*) | gacA::Tn | PMID: 16477005 | | LD1560 |
| strain, strain background (UCBPP-PA14 *P. aeruginosa*) | ΔccoN4::ccoN4; ΔN4::N4 | this study | | LD1867 |
| strain, strain background (UCBPP-PA14 *P. aeruginosa*) | ΔccoN1ΔccoN2ΔccoN4:: ccoN4; ΔN1ΔN2ΔN4::N4 | this study | | LD2576 |
| strain, strain background (UCBPP-PA14 *P. aeruginosa*) | MCS-*gfp* | this study | | LD2820 |
| strain, strain background (UCBPP-PA14 *P. aeruginosa*) | Pcco1-*gfp*; cco1Pr-*gfp* | this study | | LD2784 |
| strain, strain background (UCBPP-PA14 *P. aeruginosa*) | Pcco2-*gfp*; cco2Pr-*gfp* | this study | | LD2786 |
| strain, strain background (UCBPP-PA14 *P. aeruginosa*) | PccoN4-*gfp*; ccoN4Pr-*gfp* | this study | | LD2788 |
| strain, strain background (UCBPP-PA14 *P. aeruginosa*) | PA14-*yfp* | this study | | LD2780 |
| strain, strain background (UCBPP-PA14 *P. aeruginosa*) | ΔccoN1ΔccoN2-*yfp*; ΔN1ΔN2-*yfp* | this study | | LD2013 |
| strain, strain background (UCBPP-PA14 *P. aeruginosa*) | ΔccoN4-*yfp*; ΔN4-*yfp* | this study | | LD2834 |
| strain, strain background (UCBPP-PA14 *P. aeruginosa*) | ΔccoN1ΔccoN2Δcco N4-*yfp*; ΔN1ΔN2ΔN4-*yfp* | this study | | LD2136 |
| strain, strain background (UCBPP-PA14 *P. aeruginosa*) | Δcco1cco2-*yfp* | this study | | LD2012 |
| strain, strain background (*Escherichia coli*) | UQ950 | other | | From D. Lies, Caltech; LD44 |
| strain, strain background (*E. coli*) | BW29427 | other | | From W. Metcalf, University of Illinois; LD661 |

*Continued on next page*

*Continued*

| Reagent type (species) or resource | Designation | Source or reference | Identifiers | Additional information |
|---|---|---|---|---|
| strain, strain background (*E. coli*) | β2155 | PMID: 8990308 | | LD69 |
| strain, strain background (*E. coli*) | S17-1 | doi:10.1038/nbt1183-784 | | LD2901 |
| strain, strain background (*Saccharomyces cerevisiae*) | InvSc1 | Invitrogen | | LD676 |
| recombinant DNA reagent | pMQ30 (plasmid) | PMID: 16820502 | | For generation of deletion constructs listed above; further information can be found in the Materials and Methods section. |
| recombinant DNA reagent | pAKN69 (plasmid) | PMID: 15186351 | | For generation of strains constitutively expression *eyfp*; further information can be found in the Materials and Methods section. |
| recombinant DNA reagent | pLD2722 (plasmid) | this study | | For generation of *gfp* reporter constructs; further information can be found in the Materials and Methods section. |
| recombinant DNA reagent | pFLP2 (plasmid) | PMID: 9661666 | | For generation of *gfp* reporter constructs; further information can be found in the Materials and Methods section. |
| software, algorithm | EggNOG Database | PMID: 26582926 | | http://eggnogdb.embl.de/#/app/home |
| software, algorithm | SensorTrace Profiling | Unisense | | For data acquisition for redox and oxygen microprofiling; further information can be found in the Materials and Methods section. |
| other | Agar | Teknova | | For colony morphology assays; further information can be found in the Materials and Methods section. |
| other | Lectin | Vector Laboratories | | For visualization of matrix; further information can be found in the Materials and Methods section. |

## Bacterial strains and growth conditions

*P. aeruginosa* strain UCBPP-PA14 (*Rahme et al., 1995*) was routinely grown in lysogeny broth (LB; 1% tryptone, 1% NaCl, 0.5% yeast extract) (*Bertani, 2004*) at 37°C with shaking at 250 rpm unless otherwise indicated. Overnight cultures were grown for 12–16 hr. For genetic manipulation, strains were typically grown on LB solidified with 1.5% agar. Strains used in this study are listed in *Table 1*. In general, liquid precultures served as inocula for experiments. Overnight precultures for biological replicates were started from separate clonal source colonies on streaked agar plates. For technical replicates, a single preculture served as the source inoculum for subcultures.

## Construction of mutant *P. aeruginosa* strains

For making markerless deletion mutants in *P. aeruginosa* PA14 (*Table 1*) 1 kb of flanking sequence from each side of the target gene were amplified using the primers listed in *Table 2* and inserted into pMQ30 through gap repair cloning in *Saccharomyces cerevisiae* InvSc1 (*Shanks et al., 2006*). Each plasmid listed in *Table 3* was transformed into *Escherichia coli* strain UQ950, verified by restriction digests, and moved into PA14 using biparental conjugation. PA14 single recombinants were selected on LB agar plates containing 100 µg/ml gentamicin. Double recombinants (markerless deletions) were selected on LB without NaCl and modified to contain 10% sucrose. Genotypes of deletion mutants were confirmed by PCR. Combinatorial mutants were constructed by using single mutants as hosts for biparental conjugation, with the exception of Δcco1cco2, which was constructed by deleting the *cco1* and *cco2* operons simultaneously as one fragment. *ccoN4* complementation strains were made in the same manner, using primers LD438 and LD441 listed in *Table 2* to amplify the coding sequence of *ccoN4*, which was verified by sequencing and complemented back into the site of the deletion.

**Table 1.** Strains used in this study.

| Strain | Number | Description | Source |
|---|---|---|---|
| *Pseudomonas aeruginosa* strains | | | |
| UCBPP-PA14 | | Clinical isolate UCBPP-PA14. | *Rahme et al. (1995)* |
| PA14 Δ*phz* | LD24 | PA14 with deletions in *phzA1-G1* and *phzA2-G2* operons. | *Dietrich et al., 2006a* |
| PA14 Δ*ccoN1* | LD1784 | PA14 with deletion in PA14_44370. | this study |
| PA14 Δ*ccoN2* | LD1614 | PA14 with deletion in PA14_44340. | this study |
| PA14 Δ*ccoN3* | LD1620 | PA14 with deletion in PA14_40510. | this study |
| PA14 Δ*ccoN4* | LD2833 | PA14 with deletion in PA14_10500. | this study |
| PA14 Δ*ccoN1* Δ*ccoN2* | LD1888 | PA14 with deletions in PA14_44370 and PA14_44340. Made by mating pLD1610 into LD1784. | this study |
| PA14 Δ*ccoN1* Δ*ccoN4* | LD1951 | PA14 with deletions in PA14_44370 and PA14_10500. Made by mating pLD1264 into LD1784. | this study |
| PA14 Δ*ccoN2* Δ*ccoN4* | LD1692 | PA14 with deletions in PA14_44340 and PA14_10500. Made by mating pLD1264 into LD1614. | this study |
| PA14 Δ*ccoN3* Δ*ccoN4* | LD1649 | PA14 with deletions in PA14_40510 and PA14_10500. Made by mating pLD1264 into LD1620. | this study |
| PA14 Δ*ccoN1* Δ*ccoN2* Δ*ccoN3* | LD1977 | PA14 with deletions in PA14_443470, PA14_44340, and PA14_40510. Made by mating pLD1616 into LD1888. | this study |
| PA14 Δ*ccoN1* Δ*ccoN2* Δ*ccoN4* | LD1976 | PA14 with deletions in PA14_443470, PA14_44340, and PA14_10500. Made by mating pLD1264 into LD1888. | this study |
| PA14 Δ*ccoN1* Δ*ccoN2* Δ*ccoN4* Δ*ccoN3* | LD2020 | PA14 with deletions in PA14_443470, PA14_44340, PA14_10500, and PA14_40510. Made by mating pLD1264 into LD1977. | this study |
| PA14 Δ*cco1cco2* | LD1933 | PA14 with both *cco* operons (PA14_44340-PA14_44400) deleted simultaneously. | this study |
| PA14 Δ*cox* Δ*cyo* Δ*cio* | LD2587 | PA14 with deletions in PA14_01290–01320 (*cox*/*aa3* operon), PA14_47150–47210 (*cyo*/*bo3* operon), and PA14_13030–13040 (*cio* operon). Made by mating pLD1966, pLD1967, and pLD2044, in that order, to PA14. | this study |
| PA14 Δ*hcn* | LD2827 | PA14 with deletion in *hcnABC* operon (PA14_36310–36330). | this study |
| PA14 Δ*phz* Δ*hcn* | LD2828 | PA14 with deletions in *phzA1-G1*, *phzA2-G2*, and *hcnABC* operons. Made by mating pLD2791 into LD24. | this study |
| PA14 Δ*ccoN4* Δ*hcn* | LD2829 | PA14 with deletions in PA14_10500 and *hcnABC* operon. Made by mating pLD2791 into LD2833. | this study |
| PA14 Δ*ccoN1* Δ*ccoN2* Δ*hcn* | LD2830 | PA14 with deletions in PA14_44370, PA14_44340, and *hcnABC* operon. Made by mating pLD2791 into LD1888. | this study |
| PA14 Δ*ccoN1* Δ*ccoN2* Δ*ccoN4* Δ*hcn* | LD2831 | PA14 with deletions in PA14_44370, PA14_44340, PA14_10500 and *hcnABC* operon. Made by mating pLD2791 into LD1976. | this study |
| *Pseudomonas aeruginosa* strains | | | |
| PA14 Δ*cco1cco2* Δ*hcn* | LD2832 | PA14 with deletions in *cco1*, *cco2*, and *hcnABC* operons. Made by mating pLD2791 into LD1933. | this study |
| PA14 *gacA*::*Tn* | LD1560 | MAR2xT7 transposon insertion into PA14_30650. | *Liberati et al. (2006)* |
| PA14 Δ*ccoN4*::*ccoN4* | LD1867 | PA14 Δ*ccoN4* strain with wild-type *ccoN4* complemented back into the site of deletion. Made by mating pLD1853 into LD2833. | this study |
| PA14 Δ*ccoN1* Δ*ccoN2* Δ*ccoN4*::*ccoN4* | LD2576 | PA14 Δ*ccoN1* Δ*ccoN2* Δ*ccoN4* strain with wild-type *ccoN4* complemented back into the site of deletion. Made by mating pLD1853 into LD1976. | this study |
| PA14 MCS-gfp | LD2820 | PA14 without a promoter driving *gfp* expression. | this study |

*Table 1 continued on next page*

*Table 1 continued*

| Strain | Number | Description | Source |
|---|---|---|---|
| PA14 P*cco-1*-gfp | LD2784 | PA14 with promoter of *cco1* operon driving *gfp* expression. | this study |
| PA14 P*cco-2*-gfp | LD2786 | PA14 with promoter of *cco2* operon driving *gfp* expression. | this study |
| PA14 P*ccoN4*-gfp | LD2788 | PA14 with promoter of *ccoN4Q4* operon driving *gfp* expression. | this study |
| PA14-yfp | LD2780 | WT PA14 constitutively expressing *eyfp*. | this study |
| PA14 ΔccoN1 ΔccoN2-yfp | LD2013 | PA14 ΔccoN1 ΔccoN2 constitutively expressing *eyfp*. Made by mating pAKN69 into LD1888. | this study |
| PA14 ΔccoN4-yfp | LD2834 | PA14 ΔccoN4 constitutively expressing *eyfp*. Made by mating pAKN69 into LD2833. | this study |
| PA14 ΔccoN1 ΔccoN2 ΔccoN4-yfp | LD2136 | PA14 ΔccoN1 ΔccoN2 ΔccoN4 constitutively expressing *eyfp*. Made by mating pAKN69 into LD1976. | this study |
| PA14 Δcco1cco2-yfp | LD2012 | PA14 Δcco1cco2 constitutively expressing *eyfp*. Made by mating pAKN69 into LD1933. | this study |
| *Escherichia coli* strains | | | |
| UQ950 | LD44 | *E. coli* DH5 λpir strain for cloning. F-Δ(argF- lac) 169φ80 dlacZ58(ΔM15) glnV44(AS) rfbD1 gyrA96(NaIR) recA1 endA1 spoT thi-1 hsdR17 deoR λpir+ | D. Lies, Caltech |
| BW29427 | LD661 | Donor strain for conjugation. thrB1004 pro thi rpsL hsdS lacZ ΔM15RP4-1360 Δ(araBAD)567 ΔdapA1314::[erm pir(wt)] | W. Metcalf, University of Illinois |
| β2155 | LD69 | Helper strain. thrB1004 pro thi strA hsdsS lacZΔM15 (F'lacZΔM15 laclq traD36 proA + proB + ) ΔdapA:: erm (Ermr)pir::RP4 [::kan (Kmr) from SM10] | *Dehio and Meyer (1997)* |
| S17-1 | LD2901 | StrR, TpR, F— RP4-2-Tc::Mu aphA::Tn7 recA λpir lysogen | *Simon et al. (1983)* |
| *Saccharomyces cerevisiae* strains | | | |
| InvSc1 | LD676 | MATa/MATalpha leu2/leu2 trp1-289/trp1-289 ura3—52/ura3-52 his3-Δ1/his3-Δ1 | Invitrogen |

DOI: https://doi.org/10.7554/eLife.30205.016

## Colony biofilm morphology assays

Overnight precultures were diluted 1:100 in LB (ΔN1ΔN2, ΔN1ΔN2ΔN3, ΔN1ΔN2ΔN4, ΔN1ΔN2ΔN4ΔN3, ΔN1ΔN2ΔN4::N4, Δcco1cco2, ΔN1ΔN2Δhcn, ΔN1ΔN2ΔN4Δhcn, Δcco1cco2Δhcn, and ΔcoxΔcyoΔcio were diluted 1:50) and grown to mid-exponential phase (OD at 500 nm ≈ 0.5). Ten microliters of subcultures were spotted onto 60 ml of colony morphology medium (1% tryptone, 1% agar [Teknova (Hollister, CA) A7777] containing 40 µg/ml Congo red dye [VWR (Radnor, PA) AAAB24310-14] and 20 µg/ml Coomassie blue dye [VWR EM-3300]) in a 10 cm x 10 cm x 1.5 cm square Petri dish (LDP [Wayne, NJ] D210-16). For preparation of biofilms grown on on phenazine methosulfate (PMS), colony morphology medium was supplemented with 200 µM PMS (Amresco [Solon, OH] 0361) after autoclaving. For nitrate experiments, colony morphology medium was supplemented with 0, 10, or 40 mM potassium nitrate. Plates were incubated for up to 5 days at 25˚C with >90% humidity (Percival [Perry, IA] CU-22L) and imaged daily using a VHX-1000 digital microscope (Keyence, Japan). Images shown are representative of at least 10 biological replicates. 3D images of biofilms were taken on day 5 of development using a Keyence VR-3100 wide-area 3D measurement system. ΔcoxΔcyoΔcio, *hcn* deletion mutants, and strains grown for the nitrate experiment were imaged using a flatbed scanner (Epson [Japan] E11000XL-GA) and are representative of at least three biological replicates

## TTC reduction assay

One microliter of overnight cultures (five biological replicates), grown as described above, was spotted onto a 1% tryptone, 1.5% agar plate containing 0.001% (w/v) TTC (2,3,5-triphenyl-tetrazolium chloride [Sigma-Aldrich (St. Louis, MO) T8877]) and incubated in the dark at 25˚C for 24 hr. Spots were imaged using a scanner (Epson E11000XL-GA) and TTC reduction, normalized to colony area, was quantified using Adobe Photoshop CS5 (San Jose, CA). Colorless TTC undergoes an irreversible

**Table 2.** Primers used in this study.

| Primer number | Sequence | used to make plasmid number |
|---|---|---|
| LD717 | ccaggcaaattctgttttatcagaccgcttctgcgttctgatCAGGACAAGCAGTGGGAAC | pLD1852 |
| LD718 | aggtgttgtaggccatcagcTGGCGGACCACCTTATAGTT | |
| LD958 | aactataaggtggtccgccaCGGTGGTTTCTTCCTCACC | |
| LD959 | ggaattgtgagcggataacaatttcacacaggaaacagctGGTCCAGCCTTTTTCCTTGT | |
| LD725 | ccaggcaaattctgttttatcagaccgcttctgcgttctgatCCCCTCAGAGAAGTCAGTCG | pLD1610 |
| LD726 | aggtgttgtaggccatcaggGGCGGACCACCTTGTAGTTA | |
| LD727 | taactacaaggtggtccgccCCTGATGGCCTACAACACCT | |
| LD728 | ggaattgtgagcggataacaatttcacacaggaaacagctCAGCGGGTTGTCATACTCCT | |
| LD741 | ccaggcaaattctgttttatcagaccgcttctgcgttctgatTCGAGGGCTTCGAGAAGAT | pLD1616 |
| LD742 | aggtgttgtaggccatcagcCAGGGTCATCAGGGTGAACT | |
| LD743 | agttcaccctgatgaccctgGCTGATGGCCTACAACACCT | |
| LD744 | ggaattgtgagcggataacaatttcacacaggaaacagctCGGGTGATGTCGACGTATTC | |
| LD438 | ggaattgtgagcggataacaatttcacacaggaaacagctCCGTTGATTTCCTTCTGCAT | pLD1264 (LD438 - LD441) |
| LD439 | ctacaaggtggttcgccagtCGCTGACCTACTCCTTCGTC | pLD1853 (LD438 and LD441) |
| LD440 | gacgaaggagtaggtcagcgACTGGCGAACCACCTTGTAG | |
| LD441 | ccaggcaaattctgttttatcagaccgcttctgcgttctgatCATCGACCTGGAAGTGCTC | |
| LD725 | ccaggcaaattctgttttatcagaccgcttctgcgttctgatCCCCTCAGAGAAGTCAGTCG | pLD1929 |
| LD1063 | gttgcccaggtgttcctgtGGCGGACCACCTTGTAGTTA | |
| LD949 | ggaattgtgagcggataacaatttcacacaggaaacagctTGTAGTCGAGGGACTTCTTGC | |
| LD1064 | taactacaaggtggtccgccACAGGAACACCTGGGCAAC | |
| LD2168 | ccaggcaaattctgttttatcagaccgcttctgcgttctgatATGTAGGGATCGAGCGACAG | pLD2791 |
| LD2169 | acacgatatccagcccctctTGGACATCGCGCCGTTCCTC | |
| LD2170 | gaggaacggcgcgatgtccaAGAGGGGCTGGATATCGTGT | |
| LD2171 | ggaattgtgagcggataacaatttcacacaggaaacagctAAGAGGTCATAATCGGCGGT | |
| LD2120 | gattcgacatcactagtACGCCCAGCTCCAACAAA | pLD2777 |
| LD2121 | gattcgatgccctcgaGCTAGGGGTTCCACGGTTAAT | |
| LD2122 | gattcgactgcactagtCATCGACTTGCCGCCCAG | pLD2778 |
| LD2123 | g attcg atg ccctcgaGCTATGGGCTTCCATC CAC | |
| LD2124 | gattcgactgcactagtGGCTACTTCCTCTGGCTGG | pLD2779 |
| LD2125 | gattcgactgcctcgagCTGTACAGTCCCGAAAGAAATGAAC | |
| LD1118 | ccaggcaaattctgttttatcagaccgcttctgcgttctgatTCTTCAGGTTCTCGCGGTAG | pLD1966 |
| LD1119 | aagtgccagtaccaactggcGCAGATCCAGAAGATGGTCA | |
| LD1120 | tgaccatcttctggatctgcGCCAGTTGGTACTGGCACTT | |
| LD1121 | ggaattgtgagcggataacaatttcacacaggaaacagctATCGCGAGACTCATGGTTTT | |
| LD1134 | ccaggcaaattctgttttatcagaccgcttctgcgttctgatCGCTGCTTGTCGATCTGTT | pLD1967 |
| LD1135 | gcgacatgaccctgttcaacCTGACCGGCTACTGGACC | |
| LD1136 | ggtccagtagccggtcagGTTGAACAGGGTCATGTCGC | |
| LD1137 | ggaattgtgagcggataacaatttcacacaggaaacagctCCTCGGCGACCATGAATAC | |
| LD1126 | ccaggcaaattctgttttatcagaccgcttctgcgttctgatTTCAGGTTCTTCGGGTTCTC | pLD2044 |
| LD1187 | aacagcgcgccgaccagcatCTCTTCGTTCGTTTTCAGCC | |
| LD1188 | ggctgaaaacgaacgaagagATGCTGGTCGGCGCGCTGTT | |
| LD1189 | ggaattgtgagcggataacaatttcacacaggaaacagctGCGTTGATGAAGCGGATAAC | |

DOI: https://doi.org/10.7554/eLife.30205.017

**Table 3.** Plasmids used in this study.

| Plasmid | Description | Source |
|---------|-------------|--------|
| pMQ30 | 7.5 kb mobilizable vector; oriT, sacB, Gm$^R$. | *Shanks et al. (2006)* |
| pAKN69 | Contains mini-Tn7(Gm)PA1/04/03::eyfp fusion. | *Lambertsen et al. (2004)* |
| pLD2722 | GmR, TetR flanked by Flp recombinase target (FRT) sites to resolve out resistance casettes. | this study |
| pFLP2 | Site-specific excision vector with cI857-controlled FLP recombinase encoding sequence, sacB, Ap$^R$. | *Hoang et al. (1998)* |
| pLD1852 | Δ*ccoN1* PCR fragment introduced into pMQ30 by gap repair cloning in yeast strain InvSc1. | this study |
| pLD1610 | Δ*ccoN2* PCR fragment introduced into pMQ30 by gap repair cloning in yeast strain InvSc1. | this study |
| pLD1616 | Δ*ccoN3* PCR fragment introduced into pMQ30 by gap repair cloning in yeast strain InvSc1. | this study |
| pLD1264 | Δ*ccoN4* PCR fragment introduced into pMQ30 by gap repair cloning in yeast strain InvSc1. | this study |
| pLD1929 | Δ*cco1 cco2* PCR fragment introduced into pMQ30 by gap repair cloning in yeast strain InvSc1. | this study |
| pLD2791 | Δ*hcn* PCR fragment introduced into pMQ30 by gap repair cloning in yeast strain InvSc1. | this study |
| pLD1853 | Full genomic sequence of *ccoN4* PCR fragment introduced into pMQ30 by gap repair cloning in yeast strain InvSc1. Verified by sequencing. | this study |
| pLD1966 | Δ*aa3* PCR fragment introduced into pMQ30 by gap repair cloning in yeast strain IncSc1. | this study |
| pLD1967 | Δ*bo3* PCR fragment introduced into pMQ30 by gap repair cloning in yeast strain IncSc1. | this study |
| pLD2044 | Δ*cio* PCR fragment introduced into pMQ30 by gap repair cloning in yeast strain IncSc1. | this study |
| pLD2777 | PCR-amplified *cco1* promoter ligated into pSEK103 using SpeI and XhoI. | this study |
| pLD2778 | PCR-amplified *cco2* promoter ligated into pSEK103 using SpeI and XhoI. | this study |
| pLD2779 | PCR-amplified *ccoN4* promoter ligated into pSEK103 using SpeI and XhoI. | this study |

DOI: https://doi.org/10.7554/eLife.30205.018

color change to red when reduced. Pixels in the red color range were quantified and normalized to colony area using Photoshop CS5.

## Liquid culture growth assays

(i) Overnight precultures were diluted 1:100 (Δ*N1*Δ*N2*, Δ*N1*Δ*N2*Δ*N4*, and Δ*cco1cco2* were diluted 1:50) in 1% tryptone in a clear- flat-bottom polystyrene 96-well plate (VWR 82050–716) and grown for two hours ($OD_{500nm} \approx 0.2$). These cultures were then diluted 100-fold in 1% tryptone in a new 96-well plate and incubated at 37°C with continuous shaking on the medium setting in a Synergy 4 plate reader (BioTek, Winooski, VT). Growth was assessed by taking OD readings at 500 nm every 30 min for at least 24 hr. (ii) *hcn* mutants: Overnight precultures were diluted 1:100 (Δ*N1*Δ*N2*Δ*hcn*, Δ*N1*Δ*N2*Δ*N4*Δ*hcn*, and Δ*cco1cco2*Δ*hcn* were diluted 1:50) in MOPS minimal medium (50 mM 4-morpholinepropanesulfonic acid (pH 7.2), 43 mM NaCl, 93 mM NH$_4$Cl, 2.2 mM KH$_2$PO4, 1 mM MgSO$_4$•7H$_2$O, 1 μg/ml FeSO$_4$•7H$_2$O, 20 mM sodium succinate hexahydrate) and grown for 2.5 hr until OD at 500 nm ≈ 0.1. These cultures were then diluted 100-fold in MOPS minimal medium in a clear, flat-bottom polystyrene 96-well plate and incubated at 37°C with continuous shaking on the medium setting in a plate reader. Growth was assessed by taking OD readings at 500 nm every 30 min for at least 24 hr. (iii) Terminal oxidase reporters: Overnight precultures were grown in biological triplicate; each biological triplicate was grown in technical duplicate. Overnight precultures were diluted 1:100 in 1% tryptone and grown for 2.5 hr until OD at 500 nm ≈ 0.1. These cultures were then diluted 100-fold in 1% tryptone in a clear, flat-bottom, polystyrene black 96-well plate (VWR 82050–756) and incubated at 37°C with continuous shaking on the medium setting in a plate reader. Expression of GFP was assessed by taking fluorescence readings at excitation and emission wavelengths of 480 nm and 510 nm, respectively, every hour for 24 hr. Growth was assessed by taking OD readings at 500 nm every 30 min for 24 hr. Growth and RFU values for technical duplicates were averaged to obtain the respective values for each biological replicate. RFU values for a strain without a promoter inserted upstream of the *gfp* gene (MCS-*gfp*) were considered background and subtracted from the fluorescence values of each reporter.

## Competition assays

Overnight precultures of fluorescent (YFP-expressing) and non-fluorescent strains were diluted 1:100 in LB ($\Delta N1\Delta N2$, $\Delta N1\Delta N2\Delta N4$ and $\Delta cco1cco2$ were diluted 1:50) and grown to mid-exponential phase (OD at 500 nm $\approx$ 0.5). Exact OD at 500 nm values were read in a Spectronic 20D+ spectrophotometer (Thermo Fisher Scientific [Waltham, MA]) and cultures were adjusted to the same OD. Adjusted cultures were then mixed in a 1:1 ratio of fluorescent:non-fluorescent cells and 10 µl of this mixture were spotted onto colony morphology plates and grown for 3 days as described above. At specified time points, biofilms were collected, suspended in 1 ml of 1% tryptone, and homogenized on the 'high' setting in a bead mill homogenizer (Omni [Kennesaw, GA] Bead Ruptor 12); day 1 colonies were homogenized for 35 s while days 2 and 3 colonies were homogenized for 99 s. Homogenized cells were serially diluted and $10^{-6}$, $10^{-7}$, and $10^{-8}$ dilutions were plated onto 1% tryptone plates and grown overnight at 37°C. Fluorescent colony counts were determined by imaging plates with a Typhoon FLA7000 fluorescent scanner (GE Healthcare Life Sciences [United Kingdom]) and percentages of fluorescent vs. non-fluorescent colonies were determined.

## Construction of terminal oxidase reporters

Translational reporter constructs for the Cco1, Cco2, and CcoN4Q4 operons were constructed using primers listed in *Table 1*. Respective primers were used to amplify promoter regions (500 bp upstream of the operon of interest), adding an SpeI digest site to the 5' end of the promoter and an XhoI digest site to the 3' end of the promoter. Purified PCR products were digested and ligated into the multiple cloning site (MCS) of the pLD2722 vector, upstream of the *gfp* sequence. Plasmids were transformed into *E. coli* strain UQ950, verified by sequencing, and moved into PA14 using biparental conjugation with *E. coli* strain S17-1. PA14 single recombinants were selected on M9 minimal medium agar plates (47.8 mM $Na_2HPO_4 \cdot 7H_2O$, 22 mM $KH_2PO_4$, 8.6 mM NaCl, 18.6 mM $NH_4Cl$, 1 mM $MgSO_4$, 0.1 mM $CaCl_2$, 20 mM sodium citrate dihydrate, 1.5% agar) containing 100 µg/ml gentamicin. The plasmid backbone was resolved out of PA14 using Flp-FRT recombination by introduction of the pFLP2 plasmid (*Hoang et al., 1998*) and selected on M9 minimal medium agar plates containing 300 µg/ml carbenicillin and further on LB agar plates without NaCl and modified to contain 10% sucrose. The presence of *gfp* in the final clones was confirmed by PCR.

## Thin sectioning analyses

Two layers of 1% tryptone with 1% agar were poured to depths of 4.5 mm (bottom) and 1.5 mm (top). Overnight precultures were diluted 1:100 ($\Delta N1\Delta N2$, $\Delta N1\Delta N4$, $\Delta N1\Delta N2\Delta N4$, $\Delta cco1cco2$ were diluted 1:50) in LB and grown for 2 hr, until early-mid exponential phase. Five to 10 µl of subculture were then spotted onto the top agar layer and colonies were incubated in the dark at 25°C with >90% humidity (Percival CU-22L) and grown for up to 3 days. At specified time points to be prepared for thin sectioning, colonies were covered by a 1.5-mm-thick 1% agar layer. Colonies sandwiched between two 1.5-mm agar layers were lifted from the bottom layer and soaked for 4 hr in 50 mM L-lysine in phosphate buffered saline (PBS) (pH 7.4) at 4°C, then fixed in 4% paraformaldehyde, 50 mM L-lysine, PBS (pH 7.4) for 4 hr at 4°C, then overnight at 37°C. Fixed colonies were washed twice in PBS and dehydrated through a series of ethanol washes (25%, 50%, 70%, 95%, 3 × 100% ethanol) for 60 min each. Colonies were cleared via three 60-min incubations in Histoclear-II (National Diagnostics [Atlanta, GA] HS-202) and infiltrated with wax via two separate washes of 100% Paraplast Xtra paraffin wax (Thermo Fisher Scientific 50-276-89) for 2 hr each at 55°C, then colonies were allowed to polymerize overnight at 4°C. Tissue processing was performed using an STP120 Tissue Processor (Thermo Fisher Scientific 813150). Trimmed blocks were sectioned in 10-µm-thick sections perpendicular to the plane of the colony using an automatic microtome (Thermo Fisher Scientific 905200ER), floated onto water at 45°C, and collected onto slides. Slides were air-dried overnight, heat-fixed on a hotplate for 1 hr at 45°C, and rehydrated in the reverse order of processing. Rehydrated colonies were immediately mounted in TRIS-Buffered DAPI:Fluorogel (Thermo Fisher Scientific 50-246-93) and overlaid with a coverslip. Differential interference contrast (DIC) and fluorescent confocal images were captured using an LSM700 confocal microscope (Zeiss, Germany). Each strain was prepared in this manner in at least biological triplicates.

**Table 4.** Statistical analysis.

| *Figure 2B* | Number of values (biological replicates) | mean | median | SD | SEM | Lower 95% confidence interval of mean | Upper 95% confidence interval of mean |
|---|---|---|---|---|---|---|---|
| WT | 5 | 73.22 | 72.94 | 3.387 | 1.515 | 69.02 | 77.43 |
| ΔN4 | 5 | 68.97 | 70.6 | 6.44 | 2.88 | 60.97 | 76.96 |
| ΔN1ΔN2 | 5 | 52.18 | 50.46 | 5.142 | 2.3 | 45.79 | 58.56 |
| ΔN1ΔN4 | 5 | 11.57 | 12.42 | 2.011 | 0.8991 | 9.074 | 14.07 |
| ΔN1ΔN2ΔN4 | 5 | 0.001958 | 0.001117 | 0.001696 | 0.0007586 | −0.0001481 | 0.004064 |
| Δcco1cco2 | 5 | 0.001367 | 0.0008644 | 0.001237 | 0.0005532 | −0.0001686 | 0.002903 |
| t-test | p value | p value summary | | | | | |
| WT vs. ΔN4 | 0.2273 | ns | | | | | |
| WT vs. ΔN1ΔN2 | <0.0001 | **** | | | | | |
| WT vs. ΔN1ΔN4 | <0.0001 | **** | | | | | |
| WT vs. ΔN1ΔN2ΔN4 | <0.0001 | **** | | | | | |
| WT vs. Δcco1cco2 | <0.0001 | **** | | | | | |
| *Figure 3A* | Number of values (biological replicates) | mean | median | SD | SEM | Lower 95% confidence interval of mean | Upper 95% confidence interval of mean |
| WT-YFP | 12 | 54.95 | 54.92 | 4.387 | 1.266 | 52.16 | 57.74 |
| ΔN4-YFP | 3 | 29.92 | 30.83 | 2.234 | 1.29 | 24.37 | 35.46 |
| ΔN1ΔN2-YFP | 3 | 30.49 | 31.91 | 3.527 | 2.036 | 21.73 | 39.25 |
| ΔN1ΔN2ΔN4-YFP | 3 | 4.408 | 4.296 | 3.23 | 1.865 | −3.617 | 12.43 |
| Δcco1cco2-YFP | 3 | 7.097 | 5.306 | 4.093 | 2.363 | −3.072 | 17.27 |
| t-test | p value | p value summary | | | | | |
| WT-YFP vs. ΔN4-YFP | <0.0001 | **** | | | | | |
| WT-YFP vs. ΔN1ΔN2-YFP | <0.0001 | **** | | | | | |
| ΔN1ΔN2-YFP vs. ΔN1ΔN2ΔN4-YFP | 0.0007 | *** | | | | | |
| ΔN1ΔN2-YFP vs. Δcco1cco2-YFP | 0.0017 | ** | | | | | |
| *Figure 3—figure supplement 1A* | Number of values (biological replicates) | mean | median | SD | SEM | Lower 95% confidence interval of mean | Upper 95% confidence interval of mean |
| WT) | 12 | 45.05 | 45.08 | 4.387 | 1.266 | 42.26 | 47.84 |
| ΔN4 | 3 | 28.22 | 31.31 | 7.442 | 4.297 | 9.731 | 46.71 |
| ΔN1ΔN2 | 3 | 27.81 | 28.57 | 2.514 | 1.451 | 21.56 | 34.05 |
| ΔN1ΔN2ΔN4 | 3 | 7.002 | 6.973 | 0.7508 | 0.4335 | 5.137 | 8.867 |
| Δcco1cco2 | 3 | 5.38 | 4.183 | 2.146 | 1.239 | 0.05034 | 10.71 |
| t-test | p value | p value summary | | | | | |
| WT vs. ΔN4 | 0.0002 | *** | | | | | |
| WT vs. ΔN1ΔN2 | <0.0001 | **** | | | | | |
| ΔN1ΔN2 vs. ΔN1ΔN2ΔN4 | 0.0002 | *** | | | | | |
| ΔN1ΔN2 vs. Δcco1cco2 | 0.0003 | *** | | | | | |
| *Figure 5* | Number of values (biological replicates) | mean | median | SD | SEM | Lower 95% confidence interval of mean | Upper 95% confidence interval of mean |
| WT | 8 | 150.3 | 151.2 | 10.31 | 3.644 | 141.7 | 158.9 |
| ΔN1ΔN2 | 4 | 139.3 | 137.6 | 12.33 | 6.166 | 119.6 | 158.9 |

*Table 4 continued on next page*

*Table 4 continued*

| Figure 2B | Number of values (biological replicates) | mean | median | SD | SEM | Lower 95% confidence interval of mean | Upper 95% confidence interval of mean |
|---|---|---|---|---|---|---|---|
| ΔN4 | 7 | 131.9 | 127.8 | 8.915 | 3.369 | 123.7 | 140.2 |
| ΔN1ΔN4 | 4 | 99.96 | 99.34 | 2.726 | 1.363 | 95.62 | 104.3 |
| Δcco1cco2 | 4 | 95.19 | 95.56 | 1.559 | 0.7793 | 92.71 | 97.67 |
| ΔN1ΔN2ΔN4 | 4 | 102.8 | 99.79 | 8.664 | 4.332 | 88.98 | 116.6 |
| Δphz | 7 | 84.98 | 84.23 | 10.93 | 4.131 | 74.87 | 95.09 |
| t-test | p value | p value summary | | | | | |
| WT vs. ΔN1ΔN2 | 0.1302 | ns | | | | | |
| WT vs. ΔN4 | 0.0028 | ** | | | | | |
| WT vs. ΔN1ΔN4 | <0.0001 | **** | | | | | |
| WT vs. Δcco1cco2 | <0.0001 | **** | | | | | |
| WT vs. ΔN1ΔN2ΔN4 | <0.0001 | **** | | | | | |
| WT vs. Δphz | <0.0001 | **** | | | | | |
| Figure 6 | Number of values (biological replicates) | mean | median | SD | SEM | Lower 95% confidence interval of mean | Upper 95% confidence interval of mean |
| WT | 9 | 27.44 | 39 | 18.48 | 6.16 | 13.24 | 41.65 |
| gacA::Tn | 9 | 92.56 | 93 | 8.546 | 2.849 | 85.99 | 99.12 |
| ΔN1ΔN2 | 4 | 19 | 21.5 | 14.07 | 7.036 | −3.39 | 41.39 |
| ΔN1ΔN2ΔN4 | 6 | 64.17 | 68 | 18 | 7.35 | 45.27 | 83.06 |
| Δcco1cco2 | 9 | 70.56 | 76 | 22.69 | 7.565 | 53.11 | 88 |
| t-test | p value | p value summary | | | | | |
| WT vs. ΔN1ΔN2ΔN4 | 0.0022 | ** | | | | | |
| ΔN1ΔN2 vs. ΔN1ΔN2ΔN4 | 0.0030 | ** | | | | | |
| WT vs. ΔN1ΔN2 | 0.4362 | ns | | | | | |

DOI: https://doi.org/10.7554/eLife.30205.019

## Colony thickness measurements

Colonies were prepared for thin sectioning as described above, but growth medium was supplemented with 40 µg/ml Congo Red dye and 20 µg/ml Coomassie Blue dye. Colony height measurements were obtained from confocal DIC images using Fiji image processing software (*Schindelin et al., 2012*).

## Lectin staining

Two-day-old colonies were prepared for thin sectioning as described above. Rehydrated colonies were post-stained in 100 µg/ml fluorescein-labeled *Wisteria floribunda* lectin (Vector Laboratories (Burlingame, CA) FL-1351) in PBS before being washed twice in PBS, mounted in TRIS-buffered DAPI and overlaid with a coverslip. Fluorescent confocal images were captured using an LSM700 confocal microscope (Zeiss).

## Redox profiling of biofilms

A 25-µm-tip redox microelectrode and external reference (Unisense [Denmark] RD-25 and REF-RM) were used to measure the extracellular redox state of day 2 (~48 hr) biofilms (grown as for the colony biofilm morphology assays). The redox microelectrode measures the tendency of a sample to take up or release electrons relative to the reference electrode, which is immersed in the same medium as the one on which the sample is grown. The redox microelectrode was calibrated according to manufacturer's instructions using a two-point calibration to 1% quinhydrone in pH 4 buffer

and 1% quinhydrone in pH 7 buffer. Redox measurements were taken every 5 μm throughout the depth of the biofilm using a micromanipulator (Unisense MM33) with a measurement time of 3 s and a wait time between measurements of 5 s. Profiles were recorded using a multimeter (Unisense) and the SensorTrace Profiling software (Unisense).

### Oxygen profiling of biofilms

A 25-μm-tip oxygen microsensor (Unisense OX-25) was used to measure oxygen concentrations within biofilms during the first 2 days of development, grown as described above. For oxygen profiling on 3-day-old colonies (*Figure 4*), biofilms were grown as for the thin sectioning analyses. To calibrate the oxygen microsensor, a two-point calibration was used. The oxygen microsensor was calibrated first to atmospheric oxygen using a calibration chamber (Unisense CAL300) containing water continuously bubbled with air. The microsensor was then calibrated to a 'zero' point using an anoxic solution of water thoroughly bubbled with $N_2$; to ensure complete removal of all oxygen, $N_2$ was bubbled into the calibration chamber for a minimum of 30 min before calibrating the microsensor to the zero calibration point. Oxygen measurements were then taken throughout the depth of the biofilm using a measurement time of 3 s and a wait time between measurements of 5 s. For 6-hr-old colonies, a step size of 1 μm was used to profile through the entire colony; for 12 hr and 24 hr colonies, 2 μm; for 48 hr colonies, 5 μm. A micromanipulator (Unisense MM33) was used to move the microsensor within the biofilm and profiles were recorded using a multimeter (Unisense) and the SensorTrace Profiling software (Unisense).

### Phenazine quantification

Overnight precultures were diluted 1:10 in LB and spotted onto a 25 mm filter disk (pore size: 0.2 μm; GE Healthcare 110606) placed into the center of one 35 × 10 mm round Petri dish (VWR 25373-041). Colonies were grown for 2 days in the dark at 25°C with >90% humidity. After 2 days of growth, each colony (with filter disk) was lifted off its respective plate and weighed. Excreted phenazines were then extracted from the agar medium overnight in 5 ml of 100% methanol (in the dark, nutating at room temperature). Three hundred μl of this overnight phenazine/methanol extraction were then filtered through a 0.22 μm cellulose Spin-X column (Thermo Fisher Scientific 07-200-386) and 200 μl of the flow-through were loaded into an HPLC vial. Phenazines were quantified using high-performance liquid chromatography (Agilent [Santa Clara, CA] 1100 HPLC System) as described previously (*Dietrich et al., 2006a*; *Sakhtah et al., 2016*).

### *C. elegans* pathogenicity (slow killing) assays

Slow killing assays were performed as described previously (*Tan et al., 1999*; *Powell and Ausubel, 2008*). Briefly, 10 μl of overnight PA14 cultures (grown as described above) were spotted onto slow killing agar plates (0.3% NaCl, 0.35% Bacto-Peptone, 1 mM $CaCl_2$, 1 mM $MgSO_4$, 5 μg/ml cholesterol, 25 mM $KPO_4$, 50 μg/ml FUDR, 1.7% agar) and plates were incubated for 24 hr at 37°C followed by 48 hr at room temperature (~23°C). Larval stage 4 (L4) nematodes were picked onto the PA14-seeded plates and live/dead worms were counted for up to four days. Each plate was considered a biological replicate and had a starting sample size of 30–35 worms.

### Statistical analysis

Data analysis was performed using GraphPad Prism version 7 (GraphPad Software, La Jolla, CA). Values are expressed as mean ±SD. Statistical significance of the data presented was assessed with the two-tailed unpaired Student's t-test. Values of $p \leq 0.05$ were considered significant (*$p \leq 0.05$; **$p \leq 0.01$; ***$p \leq 0.001$; ****$p \leq 0.0001$). Full statistical reporting for relevant figures can be found in *Table 4*.

## Acknowledgements

We thank Rachel Hainline for technical assistance with competition assays, Christopher Beierschmitt for technical assistance with worm pathogenicity assays, and Konstanze Schiessl for help with image analysis and feedback on the manuscript.

## Additional information

### Funding

| Funder | Grant reference number | Author |
|---|---|---|
| National Institutes of Health | R01AI103369 | Lars EP Dietrich |
| National Science Foundation | NSF CAREER | Lars EP Dietrich |
| National Institutes of Health | training grant 5T32GM008798 | Jeanyoung Jo |

The funders had no role in study design, data collection and interpretation, or the decision to submit the work for publication.

### Author contributions

Jeanyoung Jo, Conceptualization, Resources, Data curation, Formal analysis, Funding acquisition, Validation, Investigation, Visualization, Methodology, Writing—original draft, Writing—review and editing; Krista L Cortez, Data curation, Formal analysis, Validation, Investigation, Methodology, Writing—review and editing; William Cole Cornell, Data curation, Formal analysis, Validation, Investigation, Visualization, Methodology, Writing—review and editing; Alexa Price-Whelan, Conceptualization, Formal analysis, Funding acquisition, Validation, Writing—original draft, Writing—review and editing; Lars EP Dietrich, Conceptualization, Resources, Data curation, Formal analysis, Supervision, Funding acquisition, Validation, Investigation, Visualization, Methodology, Project administration, Writing—review and editing

### Author ORCIDs

Jeanyoung Jo (iD) https://orcid.org/0000-0003-1543-1148
William Cole Cornell (iD) https://orcid.org/0000-0002-8927-1813
Alexa Price-Whelan (iD) https://orcid.org/0000-0001-7587-7534
Lars EP Dietrich (iD) https://orcid.org/0000-0003-2049-1137

### Decision letter and Author response

Decision letter https://doi.org/10.7554/eLife.30205.021
Author response https://doi.org/10.7554/eLife.30205.022

## Additional files

### Supplementary files

• Transparent reporting form
DOI: https://doi.org/10.7554/eLife.30205.020

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
