## [Decision Letter]

Thank you for sending your article entitled "An orphan *cbb_3_*-type cytochrome oxidase subunit supports *Pseudomonas aeruginosa* biofilm growth and virulence" for peer review at *eLife*. Your work has been evaluated by a Senior Editor, a Reviewing Editor and three reviewers.

Overall, the three outside reviews are supportive of publication of this contribution in *eLife*. Each of the reviewers mention the importance of this work and the overall quality of the presentation.

However, each of the reviewers has suggestions for improving the manuscript. While most of the suggestions ask for more discussion of issues in the text or changes to figures (that might not warrant new experiments), they do require significant clarification on important issues. These changes are needed either to clarify the paper for the readers or to make it clear what one can and cannot conclude form the data presented. Some examples are highlighted below.

One reviewer asks if there is additional evidence to support a direct interaction between CcoN4 and either Cco1 or Cco2. Hirai et al. 2016 indicate that *Pseudomonas* harbors 4 genes for subunit N (*ccoN1, N2, N3* and *N4*), 2 genes for subunit P (*ccoP1, P2*) and 2 genes for subunit O (*ccoO1, O2*), so it can theoretically produce 16 unique *cbb_3_* complexes (4N * 2P * 2O = 16 NPO), each containing an N subunit, a P subunit and an O subunit (www.pnas.org/cgi/doi/10.1073/pnas.1613308113). This 2016 paper claims to provide evidence that 16 different *cbb_3_* isoforms form in the *Pseudomonas* cell, i.e. all of the subunits can mix and match in the membrane to make 16 possible NPO combinations, each one a functional *cbb_3_*. If this is correct the "direct interaction between CcoN4 and either Cco1 or Cco2" asked for in the review has been demonstrated by Hirai et al., 2016, in that subunit N4 can complex with P1 and O1 from the cco1 operon and with P2 and O2 from the cco2 operon. A statement on this important issue should be included in the text.

Several reviewers raised the concern that the explanation of the *Pseudomonas cbb_3_* system will leave some readers confused. Reviewer 1 comments 1 and 2 are examples of where clarification would be beneficial. How does this system of multiple complexes work? Reviewer 1 comments 3 and 4 on the proposed reduction of phenazine by N4 make a key point for this study. In addition, reviewer 2 correctly states "the authors do need to be clear that their data are consistent with the view that the N4 subunit confers the ability to reduce phenazines – without purification of a Cco complex containing N4 there is no direct biochemical evidence that this is the case." Their hypothesis requires this test, but unless you have the data in hand it seems a stretch to require that you do this for this paper. However, it seems relatively simple to test whether the addition of a reducing agent reverses the biofilm phenotypes of the mutants (reviewer 3).

Before advising further and reaching a decision, we would like to hear your response to the concerns, along with an estimated time frame for completing any additional work and providing a point by point response to each of the comments of the reviewers.

Reviewer #1:

In An orphan *cbb_3_*-type cytochrome oxidase subunit supports *Pseudomonas aeruginosa* biofilm growth and virulence, Jo et al. establish a role for the orphan *cbb_3_*-type terminal oxidase subunit, CcoN4, in colony biofilm development. The Pseudomonads have the unique characteristic of encoding multiple orphan *cbb_3_*-type subunits, but the function of the orphan subunits has remained elusive. In this study, Jo et al., reveal the CcoN4 orphan subunit functions during colony biofilm formation, as a *ccoN4* mutant demonstrates an altered colony biofilm morphology. The authors provide genetic evidence that the altered morphology resulting from the inactivation of *ccoN4* is likely due to Cco1-CcoN4 or Cco2-CcoN4 interactions by showing that a *cco1 cco2* double mutant has an exacerbated phenotype that is phenocopied in a triple mutant inactivated for *ccoN1, ccoN2*, and *ccoN4*. The importance of CcoN4 to colony biofilm growth is further established in the result that strains harboring a functioning CcoN4 outcompete *ccoN4* mutants when oxygen becomes limited during growth in a colony biofilm growth. The phenotypes associated with ccoN4 are specific to growth in a colony biofilm because *ccoN4* mutants do not exhibit altered growth in aerated broth culture. The finding that *ccoN4* expression coincides with both *cco1* and *cco2* throughout the depth of the biofilm provides further support for the hypothesis that CcoN4 interacts with Cco1 and Cco2. *P. aeruginosa* produces cyanide, a respiratory poison that targets terminal oxidase activity and previous reports implicate CcoN4 in resistance to cyanide. However, Jo et al. demonstrate that cyanide production does not affect the *ccoN4*-dependent phenotypes. Instead, the authors provide compelling evidence that the *ccoN4* altered colony biofilm morphology is due to an impaired capacity to reduce phenazines. The amount of phenazines produced by the *cco* mutants is a not considerably different; however, Jo et al. show that reduction of phenazines is impaired in a CcoN4-dependent manner leading to the conclusion that CcoN4 is important for the reduction of phenazines. In a *Caenorhabditis elegans* model of infection, CcoN4 isoform expression of the cbb3-type cytochrome oxidase mutants is required for virulence, supporting a role for CcoN4 in *P. aeruginosa* pathogenesis. This well-reasoned and well-written manuscript would be of interest to the *eLife* readership. The following suggestions serve as potential improvements manuscript:

1) The evidence that CcoN4 functions through interaction with Cco1 or Cco2 is compelling. However, a direct Cco1-CcoN4 or Cco2-CcoN4 interaction within the colony biofilm will provide additional support for this hypothesis.

2) Alternatively, does CcoN4 contain a unique domain(s) compared to CcoN1, CcoN2 and CcoN3 that facilitates an interaction with Cco1 or Cco2? An amino acid alignment of the four CcoN subunits would help the reader visualize unique domains within CcoN4 that might be responsible for the phenotype. Consistent with this, are *ccoN4* homologues from other species that contain multiple orphan *cbb_3_*-type subunits able to complement the PA14 *ccoN4* mutant phenotype? In total, these data will refine the molecular nature of the CcoN4 subunit and add further support that interactions between CcoN4 and Cco1 or Cco2 play an important role during colony biofilm growth.

3) Does the amino sequence alignment of the CcoN subunits reveal potential phenazine interaction motifs within CcoN4? Amino acid sequence comparisons of CcoN4 proteins in Pseudomonads that harbor a *phz* homologue versus those that do not would be of particular interest.

4) The colony biofilm phenotype of the phenazine knockout is remarkably similar to the *ccoN4* mutant but considerably different compared to the *cco1 cco2* and Δ*N1* Δ*N2* Δ*N4* mutants. Jo et al. show that the thickness is the same between these mutant colony biofilms is roughly the same. The authors propose that impaired phenazine reduction is responsible for the colony biofilm of *cco1 cco2* and Δ*N1* Δ*N2* Δ*N4*. Consistent with this model, one would surmise that the *phz* mutant colony biofilm phenotype should be dominant to the *cco1 cco2* and Δ*N1* Δ*N2* Δ*N4* colony biofilm phenotypes. Simply put, how does inactivation of *phz* in the *cco1 cco2* double mutant background and the Δ*N1* Δ*N2* Δ*N4* triple mutant background affect the colony biofilm phenotype? Does adding a reducing agent reverse the phenotypes (macroscopic or thickness) of the *cco1 cco2* and Δ*N1* Δ*N2* Δ*N4* mutants?

Reviewer #2:

This manuscript makes an important contribution to understanding the apparent redundancy in *cbb_3_*subunits in *Pseudomonas aeruginosa*. The authors convincingly demonstrate that the CcoN4 subunit is critical for optimal biofilm formation and that this is a consequence of its ability to reduce phenazines. The data are clearly described in the figures and support the major conclusions. However, the authors do need to be clear that their data are consistent with the view that the N4 subunit confers the ability to reduce phenazines – without purification of a Cco complex containing N4 there is no direct biochemical evidence that this is the case.

Reviewer #3:

The authors have analyzed *Pseudomonas aeruginosa* cells that contain different versions of the *cbb_3_*-type CcO with different isoforms of subunit 1 (N). The paper contains significant information that should be published. I have worked to identify points that the authors could revise in order to help readers learn more from the data.

1) The report is focused on the *cbb_3_* oxidase containing subunit N4. However, the *cbb_3_* containing N1 seems to have nearly equal impact, while N2 and N3 have less. Restricting the conclusions to N4 limits impact, and it may be misleading. For example, in the last paragraph of the subsection “CcoN4-containing isoforms function specifically in biofilms to support community morphogenesis and respiration”, the authors state that CcON4 *cbb_3_* is used preferentially in biofilm, in comparison to CcO N1. This does not appear to be true, the two things unique to N4 and not N1 are a colony morphology and the apparent requirement for N4 to kill *C. elegans*. Otherwise the results for N1 and N4 are similar.

2) Apparently all of the deletion strains retain other types of terminal oxidases, including a *bo_3_*-type and a *bd*-type quinol oxidase plus an *aa_3_*-type cytochrome *c* oxidase. This should be discussed.

3) Colony morphology is emphasized, but the relationship of the colony to biofilm is not adequately explained. Is all of the colony biofilm? Are the ridges and the rings that project upwards in various colonies composed of cells that are growing outside of the biofilm? A diagram of a colony labeling the different zones would help. Is there any physiological explanation for the ridges and rings?

4) The requirement for *cbb_3_* oxidases for the reduction of phenazine compounds (subsection “Microelectrode-based redox profiling reveals differential phenazine reduction activity in wild-type and cco mutant biofilms”, second paragraph) is interesting but not well discussed. The explanations in the fifth paragraph of the Discussion are too vague given known CcO biochemistry. A direct interaction of phenazine with *cbb3* seems likely. Phenazines are highly unlikely to be reduced at the O2 reduction site of subunit N due to their size. However, phenazines are known to interact directly with CcOs in that these compounds are used as electron mediators to reduce cytochrome *c* when cytochrome *c* is bound to CcOs. Increased levels of reduced cytochrome *c* in the hypoxic zone may reduce phenazines at the cytochrome *c* binding site of CcOs. While the cytochrome *c* binding sites are not present on subunits N, different isoforms of N may enhance phenazine reduction at cytochrome *c* binding sites on CcOP or O. The *aa_3_*-type CcO should be capable of the same reaction.

5) The third paragraph of the Discussion is confusing. It begins with a proposal that N4 does not form heterocomplexes with the P and O subunits of CcO1 and CcO2. However, such heterocomplexes have already been demonstrated by Hirai et al., 2016.

6) In the first paragraph of the subsection “*cco* genes show differential expression across biofilm subzones”, the third and fifth paragraphs of the Discussion and elsewhere we read that *cbb_3_* containing CcON4 (and N1) are produced throughout the biofilm, especially in the hypoxic zone. In the fourth paragraph of the Discussion, the authors state "ccoN4Q4 is uniquely induced.…in the upper portion of the biofilm where O2 is available…" This apparent contradiction is confusing.

---

## [Author Response]

Reviewer #1: […] 1) The evidence that CcoN4 functions through interaction with Cco1 or Cco2 is compelling. However, a direct Cco1-CcoN4 or Cco2-CcoN4 interaction within the colony biofilm will provide additional support for this hypothesis.

We agree that direct interactions of CcoN4 with Cco1 and Cco2 are fundamental to our conclusions. Hirai et al. (2016, doi: 10.1073/pnas.1613308113) previously demonstrated formation of these heterologous Cco isoforms biochemically. We provide genetic evidence that further supports their findings and extends them to the biofilm mode of growth. While we had alluded to the formation of these complexes as shown by Hirai et al. (subsection “CcoN4-containing isoforms function specifically in biofilms to support community morphogenesis and respiration”, second paragraph, Discussion, second paragraph), we regret omitting a more detailed description and have added the appropriate text to the Introduction (fourth paragraph). These findings are also mentioned in the Results (subsection “CcoN4-containing isoforms function specifically in biofilms to support community morphogenesis and respiration”, second paragraph) and Discussion section, second paragraph.

2) Alternatively, does CcoN4 contain a unique domain(s) compared to CcoN1, CcoN2 and CcoN3 that facilitates an interaction with Cco1 or Cco2? An amino acid alignment of the four CcoN subunits would help the reader visualize unique domains within CcoN4 that might be responsible for the phenotype. Consistent with this, are ccoN4 homologues from other species that contain multiple orphan cbb_3_-type subunits able to complement the PA14 ccoN4 mutant phenotype? In total, these data will refine the molecular nature of the CcoN4 subunit and add further support that interactions between CcoN4 and Cco1 or Cco2 play an important role during colony biofilm growth.

We agree that a sequence analysis comparing the CcoN subunits could provide insight into their differential contributions to colony morphogenesis. We now include an alignment in a new supplemental figure (Figure 2—figure supplement 4), highlighting residues that are unique to CcoN4 or shared uniquely between CcoN4 and CcoN1, which showed the strongest functional redundancy with CcoN4 in our assays. To illustrate the structural arrangement of these residues, we threaded the CcoN4 sequence using the available structure of the CcoN subunit from *P. stutzeri* (Figure 2—figure supplement 4). We note in the text that “most of the highlighted residues are surface-exposed, specifically on one half of the predicted CcoN4 structure, where they may engage in binding an unknown protein partner or specific lipids. In contrast, sites that have been described as points of interaction with CcoO and CcoP are mostly conserved, further supporting the notion that CcoN4 can interact with these subunits in Cco complexes.”

We feel that this sequence analysis, combined with our genetic evidence and the biochemical evidence previously reported by Hirai et al. (2016), provide a compelling case for direct interaction between CcoN4 and the Cco1/2 O and P subunits. We agree with the reviewer that questions regarding the functions of orphan N subunits in different organisms are interesting and hope to pursue these in the future.

3) Does the amino sequence alignment of the CcoN subunits reveal potential phenazine interaction motifs within CcoN4? Amino acid sequence comparisons of CcoN4 proteins in Pseudomonads that harbor a phz homologue versus those that do not would be of particular interest.

We observe a strong phenazine reduction effect only when *ccoN*1 and *ccoN4* are both deleted (Figure 5), suggesting redundancy. Therefore, residues that are specifically shared between these subunits could be involved in mediating phenazine reduction. As outlined in our comments above, we now highlight these residues in Figure 2—figure supplement 4 and discuss this in the last paragraph of the subsection “CcoN4-containing isoforms function specifically in biofilms to support community morphogenesis and respiration”

4) The colony biofilm phenotype of the phenazine knockout is remarkably similar to the ccoN4 mutant but considerably different compared to the cco1 cco2 and ΔN1 ΔN2 ΔN4 mutants. Jo et al. show that the thickness is the same between these mutant colony biofilms is roughly the same. The authors propose that impaired phenazine reduction is responsible for the colony biofilm of cco1 cco2 and ΔN1 ΔN2 ΔN4. Consistent with this model, one would surmise that the phz mutant colony biofilm phenotype should be dominant to the cco1 cco2 and ΔN1 ΔN2 ΔN4 colony biofilm phenotypes. Simply put, how does inactivation of phz in the cco1 cco2 double mutant background and the ΔN1 ΔN2 ΔN4 triple mutant background affect the colony biofilm phenotype? Does adding a reducing agent reverse the phenotypes (macroscopic or thickness) of the cco1 cco2 and ΔN1 ΔN2 ΔN4 mutants?

The reviewer raises an important issue regarding the effects of a lack of phenazines or phenazine reduction vs. a lack of *cbb_3_*-type terminal oxidases. It has been established that Cco1 and Cco2 function as the main enzymes responsible for oxygen reduction in *P. aeruginosa* (Arai et al. (2014) doi: 10.1128/JB.02176-14). When Cco1 and Cco2 are deleted, cells not only lack the ability to reduce phenazines, as our data show, but are also defective in aerobic respiration. Furthermore, our data indicate a phenazine-independent role of the cbb_3_-type terminal oxidases and their associated CcoN subunits: while ∆*N4* and ∆*N1*∆*N2* show normal phenazine reduction (Figure 5), both of these strains exhibit fitness disadvantages compared to the wild type (Figure 3). These interwoven effects are most prominently seen in the dramatic morphological changes of ∆*cco1cco2* and ∆*N1*∆*N2*∆*N4* (Figure 2). We would not expect removal of phenazines to overcome the oxygen-dependent effects. ∆*N4* and ∆*N1*∆*N4*, for which we demonstrated phenazine reduction defects, may be the mutants in which phenazine and oxygen reduction are best separated. As this is an important issue that requires clarification, we now also discuss it in the fourth and fifth paragraphs of the Discussion.

Our prior work has shown that phenazine reduction oxidizes the cellular redox state when electron acceptors such as oxygen are not accessible (Dietrich et al., 2013). However, the enzymes mediating this phenomenon were not known. Our results suggest that *cbb_3_*-type oxidases are required for coupling oxidation of the cytoplasm to phenazine reduction. Adding reductant would decouple phenazine reduction from cellular redox balancing and therefore would not be expected to reverse the phenotypes of *cco* mutants. However, in agreement with our model, addition of an alternative electron acceptor would be expected to reverse these phenotypes (i.e., inhibit wrinkling). We have performed this experiment with nitrate, which *P. aeruginosa* is able to use as an alternative electron acceptor (Williams et al. (2007) doi: 10.1016/S0065-2911(06)52001-6). We found that nitrate inhibited wrinkling of the various *cco* mutants. These results are included as a new panel in Figure 5—figure supplement 1 and described in the last paragraph of the subsection “Microelectrode-based redox profiling reveals differential phenazine reduction activity in wild-type and *cco* mutant biofilms”.

Reviewer #2:This manuscript makes an important contribution to understanding the apparent redundancy in cbb_3_ subunits in Pseudomonas aeruginosa. The authors convincingly demonstrate that the CcoN4 subunit is critical for optimal biofilm formation and that this is a consequence of its ability to reduce phenazines. The data are clearly described in the figures and support the major conclusions. However, the authors do need to be clear that their data are consistent with the view that the N4 subunit confers the ability to reduce phenazines – without purification of a Cco complex containing N4 there is no direct biochemical evidence that this is the case.

We appreciate the reviewer’s positive comments regarding our manuscript. We agree with the reviewer’s comment that our genetic analyses support the hypothesis that the *cbb_3_*-type terminal oxidases, including the CcoN4-containing heterocomplex(es), reduce phenazines, but that biochemical data are ultimately required to directly show this enzymatic activity. We have added a statement in the main text to stress this (Discussion, sixth paragraph).

Reviewer #3:The authors have analyzed Pseudomonas aeruginosa cells that contain different versions of the cbb_3_-type CcO with different isoforms of subunit 1 (N). The paper contains significant information that should be published. I have worked to identify points that the authors could revise in order to help readers learn more from the data.1) The report is focused on the cbb_3_ oxidase containing subunit N4. However, the cbb_3_ containing N1 seems to have nearly equal impact, while N2 and N3 have less. Restricting the conclusions to N4 limits impact, and it may be misleading. For example, in the last paragraph of the subsection “CcoN4-containing isoforms function specifically in biofilms to support community morphogenesis and respiration”, the authors state that CcON4 cbb_3_ is used preferentially in biofilm, in comparison to CcO N1. This does not appear to be true, the two things unique to N4 and not N1 are a colony morphology and the apparent requirement for N4 to kill C. elegans. Otherwise the results for N1 and N4 are similar.

The reviewer is correct that our data suggest a lesser role for the CcoN2 and CcoN3 subunits in aerobically grown *P. aeruginosa* biofilms, as indicated by the various *cco* mutant phenotypes shown in Figure 2. To further demonstrate this, we made the double mutants ∆*N2*∆*N4* and ∆*N3*∆*N4* and show that these mutants phenocopy the ∆*N4* single mutant (Figure 2—figure supplement 1, subsection “CcoN4-containing isoforms function specifically in biofilms to support community morphogenesis and respiration”, second paragraph).

We agree that CcoN1 plays a significant role in the biofilm: Colony thickness and phenazine reduction are more affected in ∆*N1*∆*N4* colonies than ∆*N1* colonies. Considering that CcoN4 has no appreciable phenotype under the liquid culture conditions that we tested, we do find it noteworthy that only ∆*N4*, and not ∆*N1*∆*N2*, shows significant defects in colony thickness (Figure 5) and the single ∆*N4* mutant has a more pronounced macroscopic biofilm phenotype than even the triple deletion mutant of the other three subunits, ∆*N1*∆*N2*∆*N3* (Figure 2—figure supplement 1).

However, we acknowledge the importance of CcoN1 as our data make it clear that there is redundancy between this subunit and CcoN4. We now discuss this in the main text (subsection “CcoN4-containing isoforms function specifically in biofilms to support community morphogenesis and respiration”, second, third and last paragraphs) and also added, in response to reviewer 1, Figure 2—figure supplement 4 highlighting residues that are uniquely shared between CcoN1 and CcoN4.

2) Apparently all of the deletion strains retain other types of terminal oxidases, including a bo_3_-type and a bd-type quinol oxidase plus an aa_3_-type cytochrome c oxidase. This should be discussed.

Yes, all deletion strains still contain the other three terminal oxidases (the *aa_3_*-type, the *bo_3_*-type, and CIO) and we appreciate the reviewer raising this point. We made a triple deletion of those terminal oxidases (∆*cox*∆*cyo*∆*cio*) and added Figure 2—figure supplement 1 and Figure 5—figure supplement 1 to show that this strain has a colony morphology and a redox profile that are similar to those of the wild type and have amended the text to incorporate these results (subsection “CcoN4-containing isoforms function specifically in biofilms to support community morphogenesis and respiration”, second paragraph and subsection “Microelectrode-based redox profiling reveals differential phenazine reduction activity in wild-type and *cco* mutant biofilms”, second paragraph).

3) Colony morphology is emphasized, but the relationship of the colony to biofilm is not adequately explained. Is all of the colony biofilm? Are the ridges and the rings that project upwards in various colonies composed of cells that are growing outside of the biofilm? A diagram of a colony labeling the different zones would help. Is there any physiological explanation for the ridges and rings?

We regret that we did not explicitly describe our approach to studying redox metabolism in biofilms and appreciate the reviewer pointing this out. A biofilm is a multicellular community that produces extracellular matrix; therefore, the entirety of the colony (including all wrinkle/ridge structures) is a biofilm. The structures that form, such as wrinkles and the high ring, result from the interplay of a variety of contributing factors but are mainly determined by (i) the distribution of cells in the initial droplet pipetted onto the plate and (ii) the production of extracellular matrix, which stimulates colony wrinkling. High central rings correspond to the circumference of the initial droplet and can be attributed in part to the “coffee ring effect” (Sempels et al. 2013). We have shown that matrix production is regulated in part by redox-sensing proteins that modulate intracellular signaling pathways (Okegbe et al. 2017), and suspect that the mutant phenotypes reported in this study could be modulated in this way. We have alluded to our work using the colony (biofilm) morphology assay to study redox metabolism and redox-driven community development in the fifth paragraph of the Introduction, the first paragraph of the subsection “CcoN4-containing isoforms function specifically in biofilms to support community morphogenesis and respiration” and the second paragraph of the Discussion.

4) The requirement for cbb_3_ oxidases for the reduction of phenazine compounds (subsection “Microelectrode-based redox profiling reveals differential phenazine reduction activity in wild-type and cco mutant biofilms”, second paragraph) is interesting but not well discussed. The explanations in the fifth paragraph of the Discussion are too vague given known CcO biochemistry. A direct interaction of phenazine with cbb3 seems likely. Phenazines are highly unlikely to be reduced at the O2 reduction site of subunit N due to their size. However, phenazines are known to interact directly with CcOs in that these compounds are used as electron mediators to reduce cytochrome c when cytochrome c is bound to CcOs. Increased levels of reduced cytochrome c in the hypoxic zone may reduce phenazines at the cytochrome c binding site of CcOs. While the cytochrome c binding sites are not present on subunits N, different isoforms of N may enhance phenazine reduction at cytochrome c binding sites on CcOP or O. The aa_3_-type CcO should be capable of the same reaction.

We thank the reviewer for raising this interesting point. The fact that phenazines are popular electron donors for biochemical studies of cytochrome oxidases does imply that these compounds can interact with the complex. We agree that different isoforms of N may indirectly enhance phenazine reduction at cytochrome c binding sites on CcoO or CcoP by affecting the overall structure of the complex. Furthermore, earlier reports have suggested that the cytochrome *bc_1_* complex is the site of phenazine reduction in isolated mitochondria (Armstrong and Stewart-Tull (1971) doi: 10.1099/00222615-4-2-263); if this is the case in *P. aeruginosa*, alterations in *cbb_3_* oxidase function may indirectly affect activity at other sites in the respiratory chain. While we have observed that phenazine reduction is influenced by the specific N subunit present in the *cbb_3_* oxidase isoform, at this stage we don’t feel that we can speculate as to the exact site of phenazine reduction. We have added these points to the Discussion section (sixth paragraph).

5) The third paragraph of the Discussion is confusing. It begins with a proposal that N4 does not form heterocomplexes with the P and O subunits of CcO1 and CcO2. However, such heterocomplexes have already been demonstrated by Hirai et al., 2016.

We agree that the wording of this paragraph was confusing and we have revised it to more accurately describe our results in the context of those reported in Hirai et al. (2016) (Discussion, third paragraph).

6) In the first paragraph of the subsection “cco genes show differential expression across biofilm subzones”, the third and fifth paragraphs of the Discussion and elsewhere we read that cbb_3_ containing CcON4 (and N1) are produced throughout the biofilm, especially in the hypoxic zone. In the fourth paragraph of the Discussion, the authors state "ccoN4Q4 is uniquely induced.…in the upper portion of the biofilm where O2 is available…" This apparent contradiction is confusing.

Expression data from our reporter constructs show that CcoN4 is produced throughout the biofilm (Figure 4). In light of these results, the line pointed out by reviewer 3 does seem to be contradictory and warrants clarification. Our intent was to highlight that CcoN4 had been shown previously to be upregulated under low oxygen conditions in liquid culture and therefore its expression in well-oxygenated regions may be a biofilm-specific expression pattern. We have revised the text to better explain our conclusions (Discussion, fourth paragraph).